# The E3 ubiquitin ligase HectD3 attenuates cardiac hypertrophy and inflammation in mice

Ashraf Yusuf Rangrez [1,2 ✉], Ankush Borlepawar[1,2,7], Nesrin Schmiedel[1,2,7], Anushka Deshpande[1,2], Anca Remes[1,2], Manju Kumari[3], Alexander Bernt[1,2], Lynn Christen[1], Andreas Helbig[4], Andreas Jungmann[5], Samuel Sossalla[6], Andreas Tholey[4], Oliver J. Müller [1,2], Derk Frank[1,2] & Norbert Frey[1,2 ✉]

Myocardial inflammation has recently been recognized as a distinct feature of cardiac hypertrophy and heart failure. HectD3, a HECT domain containing E3 ubiquitin ligase has previously been investigated in the host defense against infections as well as neuroinflammation; its cardiac function however is still unknown. Here we show that HectD3 simultaneously attenuates Calcineurin-NFAT driven cardiomyocyte hypertrophy and the pro-inflammatory actions of LPS/interferon-γ via its cardiac substrates SUMO2 and Stat1, respectively. AAV9-mediated over-expression of HectD3 in mice in vivo not only reduced cardiac SUMO2/Stat1 levels and pathological hypertrophy but also largely abolished macrophage infiltration and fibrosis induced by pressure overload. Taken together, we describe a novel cardioprotective mechanism involving the ubiquitin ligase HectD3, which links anti-hypertrophic and anti-inflammatory effects via dual regulation of SUMO2 and Stat1. In a broader perspective, these findings support the notion that cardiomyocyte growth and inflammation are more intertwined than previously anticipated.

[1] Department of Internal Medicine III (Cardiology, Angiology, Intensive Care), University Medical Center Kiel, Kiel, Germany. [2] DZHK (German Centre for Cardiovascular Research), partner site Hamburg/Kiel/Lübeck, Kiel, Germany. [3] Department of Biochemistry, University Medical Centre Hamburg-Eppendorf, Hamburg, Germany. [4] Institute for Experimental Medicine – AG Proteomics & Bioanalytics, Christian-Albrechts University Kiel, Kiel, Germany. [5] Department of Internal Medicine III, University of Heidelberg, Heidelberg, Germany. [6] Department of Internal Medicine II, University Medical Center Regensburg, Regensburg, Germany. [7] These authors contributed equally: Ankush Borlepawar, Nesrin Schmiedel. ✉email: ashraf.rangrez@uksh.de; norbert.frey@uksh.de

Myocardial hypertrophy is considered to be an adaptive response to hemodynamic stress which can be reversible and compensatory e.g. in case of physiological stimuli such as pregnancy or physical activity[1,2]. Chronic biomechanical stress however can lead to pathological or maladaptive hypertrophy characterized by cardiac remodeling, myocardial dysfunction, and fibrosis, which may progress to end-stage heart failure[3,4]. A calcineurin-dependent transcriptional pathway is one of the key signaling pathways associated with cardiac hypertrophy[5]. We recently identified small ubiquitin-like modifier 2 (SUMO2) as a potent inducer of calcineurin-NFAT signaling through an unbiased screen[6]. SUMO2 is one of the members of a small family of proteins that mediate a post-translational modification (PTM) called sumoylation, where the diglycine motif of SUMO proteins is covalently attached to its target protein via a lysine residue[7,8]. Recent advancements in the field of proteomics helped to identify novel SUMO targets, such as SERCA2A, ERK5, SIRT1, PPARα PPARγ, etc. (reviewed in ref. [9]). Subsequent analyses demonstrated that sumoylation is a dynamic process essential for the regulation of diverse cellular functions including enzymatic activity, stability, localization or degradation, protein–protein interactions, transcription, cell division, chromatin modification, etc. It is thus not surprising that alterations in sumoylation of specific protein targets have been linked to human diseases including cardiomyopathy, cardiac hypertrophy, and heart failure[7,10–13].

Inflammation including activation of proinflammatory signaling pathways, increased secretion of cytokines and chemokines, interstitial fibrosis and infiltration of inflammatory cells, e.g. macrophages, is an only recently recognized distinct hallmark of cardiac hypertrophy[14–17]. While it is still unclear whether myocardial inflammation is a cause or a consequence of cardiac hypertrophy, and the exact role inflammation plays in hypertrophy is elusive, it is largely believed that inflammation exacerbates the disease condition. Recently, SUMO2/3 have been implicated in inflammatory pathways as they negatively regulate the interferon response via sumoylation of key signaling biomolecules such as Stat1[18,19]. Stat1, the founding member of the STAT protein family, is essential for target gene activation in response to interferon stimulation[20]. Specifically, phosphorylation of Stat1 at Tyr-701 is required for its activation and nuclear translocation, dimerization, DNA binding, and specific target gene activation[21,22]. Conversely, sumoylation of Stat1 by SUMO2/3 has been shown to inhibit its phosphorylation thereby inhibiting interferon signaling[19].

To the best of our knowledge, here we identified SUMO2 and signal transducer and activator of transcription-1 (Stat1) as novel cardiac substrates for HectD3. The aim of the present study was thus to decipher the potential cardioprotective role of HectD3 using in vitro and in vivo models of cardiac pathophysiology. Interestingly, HectD3 overexpression not only attenuated SUMO2–calcineurin–NFAT signaling driven cardiomyocyte hypertrophy but also abrogated the pro-inflammatory effects of lipopolysaccharide (LPS) or interferon-γ in cultured cardiomyocytes in vitro. Consistently, AAV9-mediated overexpression of HectD3 in mice reduced cardiac SUMO2/Stat1 levels thereby alleviating pathological hypertrophy and macrophage infiltration induced by pressure overload. The beneficial effects exerted by AAV-mediated overexpression of HectD3 reveal a translational perspective and support further exploration of its therapeutic potential in cardiac hypertrophy and inflammation.

## Results
### HectD3 directly interacts with SUMO2 and suppresses activation of calcineurin-signaling and cardiomyocyte hypertrophy.
Through an unbiased NFAT-reporter-driven luciferase activity assay screen, we recently identified SUMO2 as a robust activator of calcineurin signaling[6]. We then performed a Yeast two-hybrid screen in the quest of identifying cardiac interactors of SUMO2, which identified HectD3 as a novel putative-binding partners of SUMO2 through binding of its APC10 domain (Fig. 1a). This interaction was further verified and its physiological significance was determined by performing native immunoprecipitation (IP) and co-immunostaining of HectD3 with SUMO2, using mouse heart protein lysate or neonatal rat ventricular cardiomyocytes (NRVCMs), respectively (Fig. 1b, c). Using quantitative real-time PCR, we found that HectD3 is expressed ubiquitously and at comparable levels in the heart, brain, kidney, and muscle (Supplementary Fig. 1A) pointing towards broader cellular role for HectD3. Interestingly, HectD3 overexpression attenuated SUMO2-mediated or calcineurin A (CnA)-mediated induction of NFAT-response-element-driven firefly luciferase activity (Fig. 1d). In contrast, HectD3 knockdown led to significantly higher induction of luciferase activity compared to the respective controls (Supplementary Fig. 1B). The inhibitory effects of HectD3 overexpression on SUMO2-mediated or CnA-mediated calcineurin signaling were also reflected on cardiomyocyte hypertrophy as evidenced by a reduced cell surface area and significant downregulation of natriuretic peptides A and B (nppa, nppb, respectively) (Fig. 1e, f and Supplementary Fig. 1C, D), while knockdown of HectD3 exhibited neutral effects on cardiomyocyte hypertrophy (Supplementary Fig. 1E, F).

Our previous findings suggested a novel, sumoylation-independent mechanism of SUMO2-dependent induction of calcineurin/NFAT signaling by targeting calcineurin to the nucleus[6]. Thus, we next asked whether and how the effects of HectD3 on SUMO2 are translated downstream to calcineurin/NFAT. To answer this, we performed subcellular fractionation of NRVCM cell lysates overexpressing different combinations of constitutively active CnA (δCnA), SUMO2, and HectD3 (Fig. 1g). HectD3 overexpression led to reduced nuclear localization of both CnA and δCnA (Fig. 1g (left panel), 1h). We also found a moderate nuclear localization of HectD3 (Fig. 1g (left panel)). Strikingly, SUMO2 and its sumoylated substrate protein levels were significantly reduced in the cytosol as well as in the nucleus of cardiomyocytes where HectD3 was overexpressed (Fig. 1g (right panel), 1h).

### HectD3 regulates SUMO2-mediated protein sumoylation.
Due to the fact that HectD3 is an E3 ubiquitin ligase, and based on our observation that HectD3 overexpression led to reduced levels of both cytosolic and nuclear levels of SUMO2, we hypothesized that HectD3 affects SUMO2-mediated sumoylation in NRVCMs. Overexpression of HectD3 indeed significantly reduced endogenous as well as overexpressed protein levels of SUMO2 and SUMO2-mediated sumoylation (Fig. 2a–c). Similar results were obtained in HEK293A cells when HectD3 was overexpressed in the absence or in the presence of SUMO2 (Supplementary Fig. 2A–C). We also determined if HectD3 affects SUMO1, which however was unaltered by the overexpression of HectD3 in NRVCMs (Supplementary Fig. 2D–G). Interestingly, an inverse correlation of expression levels of SUMO2 and HectD3 was observed in human heart samples from hypertrophic cardiomyopathy patients (Fig. 2d, f). In congruence with these results, HectD3 was found to be significantly downregulated at transcript level in mouse heart after pressure overload; at protein level however, though we observed a trend of downregulation, no significance was attained (Supplementary Fig. 2H–J). Furthermore, to determine if the observed effects of HectD3 on SUMO2 are UPS dependent, we treated NRVCMs with MG132, a potent, reversible, and cell-permeable proteasome inhibitor that results in the accumulation of ubiquitin-conjugated substrate proteins.

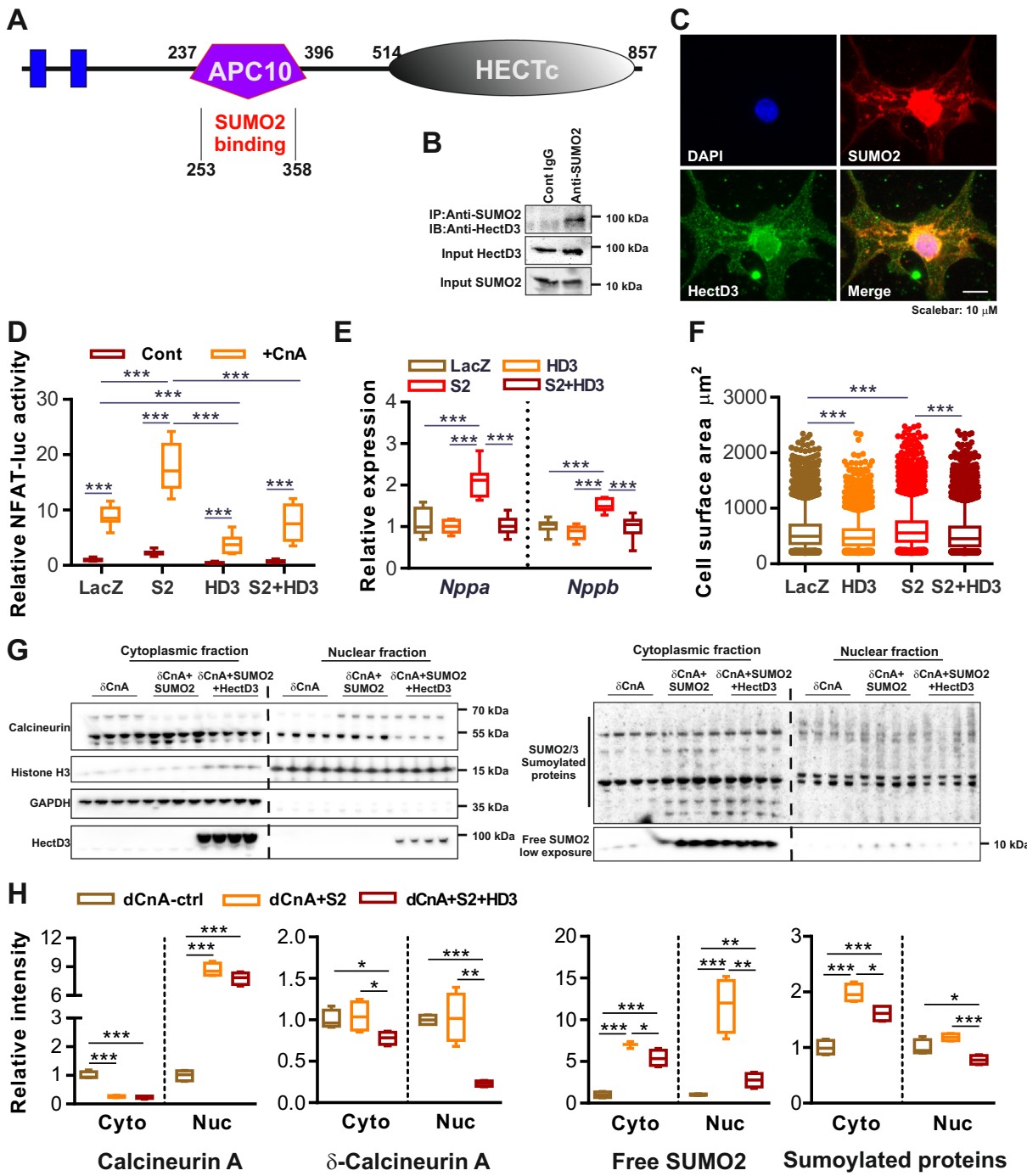

MG132 treatment attenuated SUMO2 degradation even when HectD3 was overexpressed, supporting the functional importance of this interaction (Fig. 2e, g). Consistently, immunoprecipitation assays indicated accelerated ubiquitination of SUMO2 proteins in the presence of HectD3 (Supplementary Fig. 2K, L). In contrast, knockdown of HectD3 in NRVCMs resulted in accumulation of SUMO2-sumoylated proteins (Supplementary Fig. 2M–O). Taken together, these results indicate that HectD3 regulates SUMO2 and its substrate proteins via the ubiquitin-proteasome system.

**HectD3 targets interferon response proteins and Stat1 signaling in NRVCMs.** Since no cardiac targets of HectD3 are known yet, we

employed an untargeted TMT labeling proteomics approach using liquid chromatography tandem–mass spectrometry (LC–MS/MS) to identify putative HectD3 substrates in NRVCMs. In total, we identified 5001 proteins that displayed quantifiable TMT-reporter ion signals, of which 4013 proteins were detected and quantified in both control (LacZ overexpressing) and HectD3-overexpressing conditions. Several of these proteins were found to be differentially regulated in NRVCMs upon HectD3 overexpression (Fig. 3a). Remarkably, many proteins from the interferon response signaling pathway, including Stat1 (signal transducer and activator of transcription 1), displayed reduced levels upon HectD3 overexpression (Fig. 3b). Since Stat1 is a known transcription factor that regulates expression of several downstream targets, we performed

**Fig. 1 HectD3 interacts with and suppresses SUMO2-dependent activation of calcineurin-signaling and cardiomyocyte hypertrophy. a** Pictorial representation of functional domains of HectD3 and its interaction region with SUMO2. The SUMO2-binding region from Yeast-two hybrid assay was located in the APC10 domain of HectD3 which is also involved in the E3 ubiquitin ligation of its target proteins. **b** The SUMO2–HectD3 interaction was further confirmed by immunoprecipitation of the endogenous proteins using mouse heart lysate. SUMO2-interacting proteins were pulled-down using anti-SUMO2/3 antibody and were evaluated by immunoblotting using anti-HectD3 antibody. **c** Immunofluorescence microscopy suggests scattered localization of HectD3 in cytoplasm and perinuclear region where it co-localizes with SUMO2. **d** NFAT response element-driven Firefly luciferase reporter assay was performed in NRVCMs to study the effect of overexpression of HectD3 (HD3) in and/or SUMO2 (S2), with or without constitutively active calcineurin A (CnA). Box plot indicates the inhibitory effects of HectD3 on SUMO2-mediated or CnA-mediated activation of luciferase activity ($n = 12$ each). **e** Expression of fetal genes *nppa* and *nppb* determined by quantitative real-time PCR indicates downregulation of both the genes when HectD3 is overexpressed ($n = 9$ each). **f** Cell surface area measurement of NRVCM overexpressing SUMO2, HectD3, or both, indicates SUMO2 increases cellular hypertrophy, whereas, the presence of HectD3 attenuated this activation ($n > 1500$ each). **g** Immunoblots showing the expression of Calcineurin A (CnA), SUMO2/3, and HectD3 of cytosolic or nuclear protein extracted from neonatal rat cardiomyocytes expressing delta calcineurin A (δCnA) alone, with SUMO2, or with SUMO2 and HectD3. Histone H3 and GAPDH were used as markers for nuclear or cytoplasmic proteins, respectively. **h** Densitometry analysis of CnA indicates its reduced or increased expression in cytosol or nucleus, respectively, upon SUMO2 overexpression; the presence of HectD3 displayed no effect ($n = 4$ each). Densitometry analysis of δCnA however indicates its significantly reduced expression in both cytosol and nucleus in cardiomyocytes, where HectD3 was overexpressed together with SUMO2 and δCnA. Densitometry analysis of free SUMO2/3 indicates its significantly reduced expression in both cytosol and nucleus of cardiomyocytes where HectD3 was overexpressed together with SUMO2 and δCnA ($n = 4$ each). Statistical calculations were carried out by one-way ANOVA. *$p < 0.05$, **$p < 0.01$, ***$p < 0.001$.

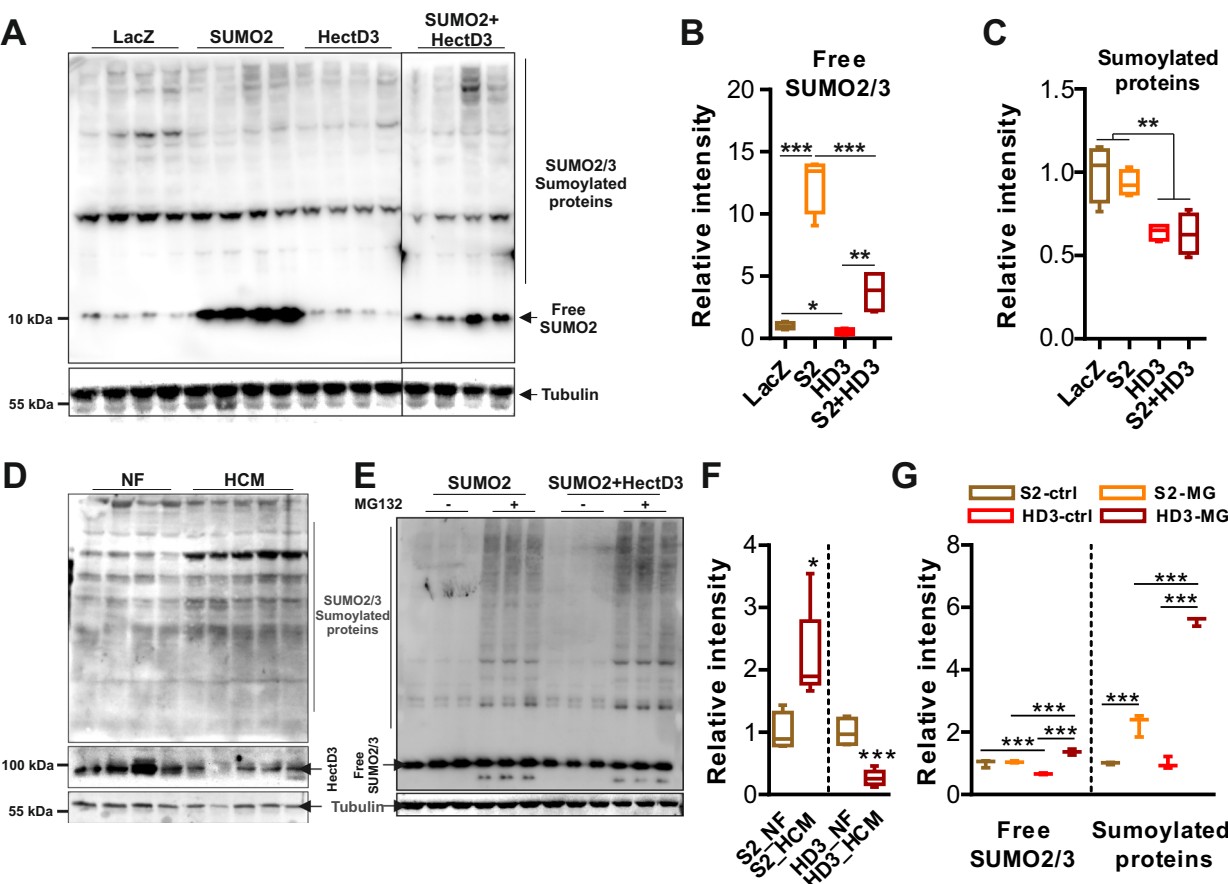

**Fig. 2 HectD3 regulates SUMO2-mediated sumoylation. a** Immunoblot indicating the expression of SUMO2/3 and sumoylated proteins in neonatal rat cardiomyocytes (NRVCMs) overexpressing SUMO2, HectD3, both, or LacZ. **b** and **c** represents densitometry analysis of immunoblot **a** for free SUMO2/3 or sumoylated proteins, respectively ($n = 4$ each). Adenovirus-mediated expression of HectD3 in NRVCMs is found to downregulate SUMO2/3 and sumoylated proteins, both endogenous as well as overexpressed. NRVCMs expressing LacZ were used as a control condition. **d** Immunoblot indicating the expression of HectD3 and SUMO2/3-sumoylated proteins in cardiac protein lysates obtained from human patients suffering from hypertrophic cardiomyopathy (HCM). Non-failing (NF) heart samples were used as controls. **e** Immunoblot indicating the expression of SUMO2/3 and sumoylated proteins in neonatal rat cardiomyocytes (NRVCMs) overexpressing SUMO2 or both, HectD3 and SUMO2, in the absence or the presence of MG132, a proteasome inhibitor. **f** Box plot representing densitometry analysis of SUMO2/3-sumoylated proteins and HectD3 shown in image **d** ($n = 4$ (control), 5 (HCM)). HectD3 is found to be significantly downregulated, whereas, SUMO2/3-mediated protein sumoylation is significantly increased in HCM patient hearts. **g** Box plot representing densitometry analysis of free SUMO2/3 or sumoylated proteins shown in image **e** clearly shows that MG132 attenuates degradation of SUMO2/3 or sumoylated proteins by HectD3 ($n = 3$). Statistical calculations were carried out by one-way ANOVA. *$p < 0.05$, **$p < 0.01$, ***$p < 0.001$.

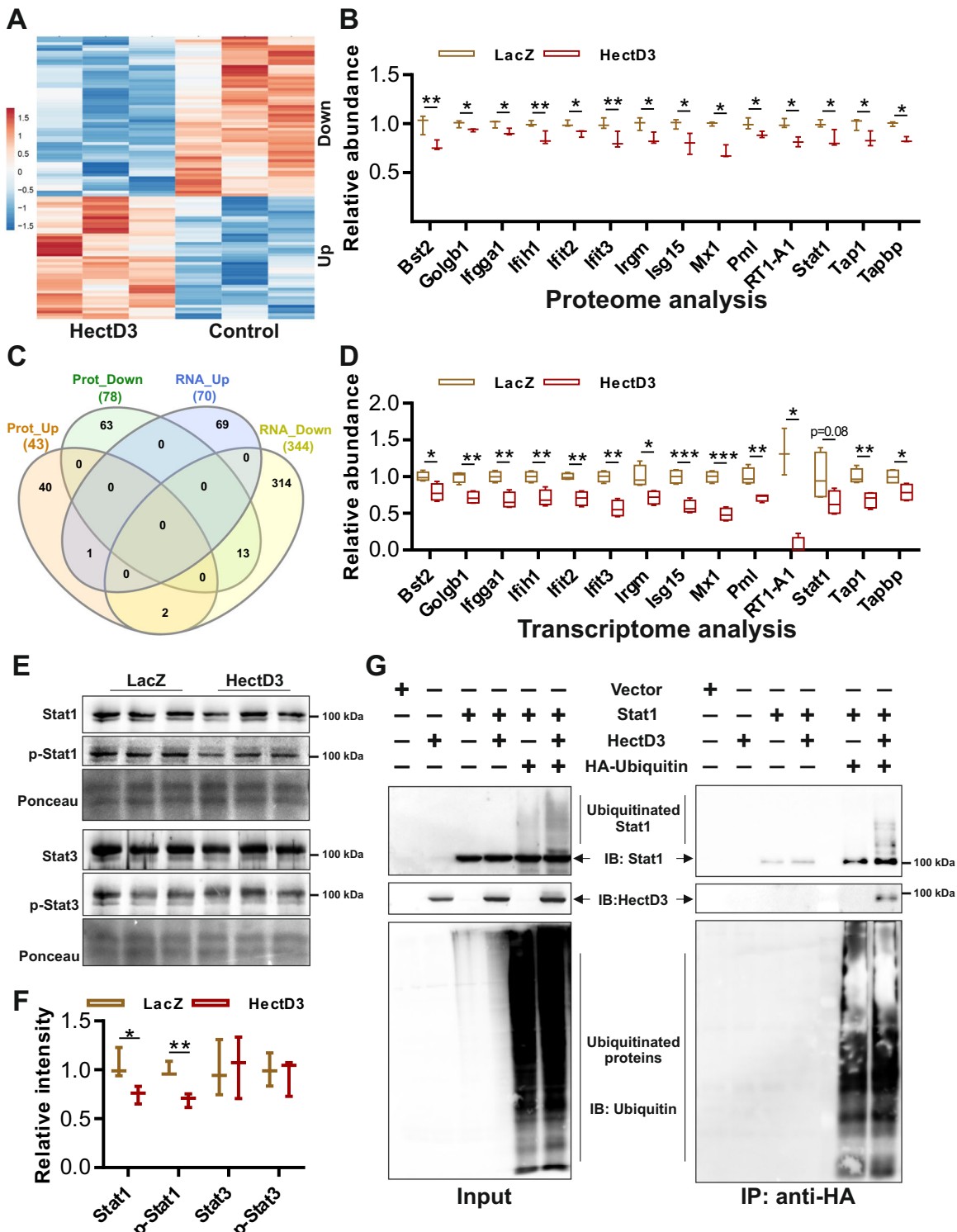

transcriptional profiling of NRVCMs overexpressing HectD3 in comparison with control cells (expressing LacZ). In congruence with the proteomics data, RNA-seq analyses revealed significant down-regulation of several direct transcriptional targets of Stat1, including numerous inflammatory response markers (Supplementary Fig. 3A–C). A meta-analysis of the proteomics and RNA-seq data revealed 13 common molecules that were down-regulated by HectD3 over-expression, six of which (Bst2, Ccl2, Mx1, Pml, STAT1 itself, and

Tap1) are known targets of Stat1[23], whereas, the remaining genes are targets of other transcription factors involved in interferon respon-sive signaling, emphasizing the notion of broad inhibition of Stat1-mediated transcriptional activation by HectD3 (Fig. 3b–d).

While further dissecting the HectD3–Stat1 interaction, we observed that HectD3 reduced the activation of Stat1 by attenuating its phosphorylation (Fig. 3e, f). In contrast, Stat3 levels or its activation were unaffected by HectD3 overexpression

**Fig. 3 HectD3 targets interferon response proteins and Stat1 signaling in NRVCMs. a** Heat map showing the differentially abundant proteins in NRVCMs after HectD3 overexpression quantified by LC–MS/MS. Protein lysate from NRVCMs overexpressing LacZ ($n = 3$ each). **b** A number of proteins involved in interferon response signaling, including Stat1, displayed significantly reduced levels during HectD3 overexpression. **c** Venn-diagram indicating the comparative analysis of differentially expressed proteins and mRNAs determined by mass spectrometry or RNA-sequencing, respectively. Overall, 13 molecules were significantly downregulated, both, at protein and RNA levels, which are shown in box plots **b** and **d**, respectively. **e** Immunoblot image displaying the expression of Stat1/p-Stat1 and Stat3/p-Stat3 in NRVCMs overexpressing LacZ (as a control) or HectD3. **f** Stat3 and p-Stat3 were not significantly altered, whereas, Stat1 and p-Stat1 were found to be significantly downregulated upon HectD3 overexpression as determined by densitometry analysis of image shown in **e** ($n = 3$ each). **g** Immunoblots indicating the input and IP samples showing Stat1, HectD3, and ubiquitin detected by anti-Stat1, anti-HectD3, and anti-HA antibodies, respectively. Images show that anti-HA beads could successfully pull down HectD3 as well as polyubiquitinated Stat1 only in the condition where HectD3 and ubiquitin were expressed with Stat1. Statistical calculations were carried out by two-tailed Student's $t$-test. $*p < 0.05$, $**p < 0.01$, $***p < 0.001$.

(Fig. 3e, f). Notably, SUMO2/3 has earlier been reported to inhibit Stat1 activation/phosphorylation via its sumoylation[19]. In NRVCMs however, SUMO2 overexpression boosted Stat1 phosphorylation, which was again attenuated by HectD3 overexpression (Supplementary Fig. 3D–F). Moreover, immunoprecipitation experiments and in vitro ubiquitination assay confirmed poly-ubiquitination of Stat1 indicating it as a novel bona fide target of HectD3 (Fig. 3g, Supplementary Fig. 3G). Altogether, these data suggest a dual regulation of Stat1 signaling by HectD3, directly via targeting Stat1 and indirectly via regulating SUMO2.

**HectD3 inhibits LPS-/IFNγ-mediated activation of the inflammatory response in cardiomyocytes.** As HectD3 appears to inhibit Stat1-signaling, we investigated if HectD3 is also sufficient to counter the proinflammatory effects of LPS and interferon γ (IFNγ) via regulating Stat1. As anticipated, LPS/IFNγ treatments significantly induced Stat1 phosphorylation, yet, HectD3 significantly hindered this activation (Fig. 4a–f). Interestingly, LPS stimulated the activation of Stat3 while no effect was observed with IFNγ, and HectD3 further reduced this activation (Supplementary Fig. 4A–F). Interferon responsive genes like bst2, ifit2, ifit3, mx1, stat1, and stat3 were strongly upregulated by LPS treatment on transcript level, while, HectD3 overexpression again significantly reduced the expression of these genes, indicating that HectD3 upregulation is indeed sufficient to inhibit interferon/Stat1-signaling in NRVCMs (Fig. 4g). Conversely, Stat1/Stat3 levels (and activation) as well as expression of downstream interferon response genes were significantly increased when HectD3 was knocked-down in NRVCMs (Supplementary Fig. 4G–I).

**HectD3 suppresses SUMO2-dependent activation of calcineurin-signaling and cardiac hypertrophy in vivo.** In order to investigate these promising in vitro results also in vivo, we used an AAV-mediated overexpression approach for heart-targeted and continuous but controlled overexpression of HectD3 in mouse models of pressure overload. AAV9-HectD3 vector, harboring the HectD3 cDNA under control of the cardiac troponin T promoter[24], was thus intravenously injected at different doses (control, $1 \times 10^{10}$, $1 \times 10^{11}$, $1 \times 10^{12}$/mouse) in mouse tail veins in order to define the appropriate dose for overexpression (Fig. 5a). A vector dose of $1 \times 10^{12}$ led to HectD3 overexpression of ~5-fold at RNA and ~2.5-fold at protein level specifically in the heart (Fig. 5b–d). Increased HectD3 expression levels did not alter basal heart weight:body weight ratios after 4 weeks of injection (Fig. 5e). We then performed transverse aortic constriction (TAC) 4 weeks post AAV injections as shown in Fig. 5f. TAC robustly induced cardiac and cellular hypertrophy (Fig. 5g, h) and worsened cardiac systolic function as shown by a reduction in fractional shortening (Fig. 5i). Consistent with the in vitro findings, overexpression of HectD3 in mouse hearts inhibited cardiac hypertrophy and moderately improved

cardiac function compared to control AAV-injected mice after TAC (Fig. 5g–i). In line with earlier reports[6,25], TAC led to upregulation of SUMO2 and its sumoylated protein levels (Supplementary Fig. 5A–C). HectD3 overexpression however markedly reduced the levels of free SUMO2/3 and its sumoylated substrates (Supplementary Fig. 5A–C). HectD3 also significantly reduced the activation of fetal/hypertrophy marker genes (nppa, nppb, rcan1.4, and myh7, while myh6 expression was moderately increased in the presence of HectD3) (Fig. 5j–n), as well as expression of profibrotic genes (col1a, col3a) (Fig. 5o, p). Similar but relatively milder effects were observed in mice where hypertrophy was induced by Angiotensin-II (AngII) treatment using osmotic minipumps (Supplementary Fig. 5D–K).

**HectD3 attenuates the inflammatory response due to pathological hypertrophy in vivo.** Our in vitro data suggested that HectD3 blunts the inflammatory effects of LPS/IFNγ via inhibition of Stat1 signaling. We thus analyzed Stat1 expression status in AAV9-HectD3-injected mice at baseline or after TAC to confirm the mechanistic in vitro findings and the observed protective phenotype of HectD3 in vivo (Fig. 6a). We discovered significantly reduced levels of activated Stat1/Stat3 following HectD3 overexpression in mouse hearts at baseline (Fig. 6b and c). TAC led to strong activation of Stat1 and inhibition of Stat3 in control mice, while HectD3 overexpression again revealed opposite effects (Fig. 6d, e). Of note, Stat3 activation was severely reduced after TAC in control mice. Overexpression of HectD3, however, retained its activation state to a level comparable to sham-operated control mice (Fig. 6d, e). Since Stat1 activation was found to be alleviated by HectD3 in mouse hearts after TAC, we anticipated a reduction in the inflammatory response. In addition to stat1, transcript levels of several interferon response genes and Stat1 signaling targets such as bst2, ccl2, ccl5, ifit1, ifit3, and ifnγ were markedly increased in control mouse hearts after TAC (Fig. 6f). HectD3 overexpression, however, potently reduced upregulation of these genes after pressure overload due to TAC (Fig. 6f). Interestingly, stat3 transcript levels remained unchanged in either of the conditions (Fig. 6f). Consistent results were also observed in AngII-treated mice compared to the respective control group (Supplementary Fig. 6A–H).

Macrophage proliferation and its recruitment into cardiac tissue contributes to tissue damage by inducing pathologic tissue remodeling and thus modulates the transition to heart failure[26]. Since Stat1 activation was shown to enhance macrophage migration[27], we next examined whether cardiac HectD3 overexpression influences immune cell infiltration in myocardial hypertrophy models. Immunohistochemical analysis indeed revealed a striking accumulation of F4/80-positive cells into the mouse myocardium following TAC and AngII treatment (Fig. 6g, h). In contrast, mice receiving AAV9-HectD3 gene therapy prior to pro-hypertrophic treatment revealed a marked

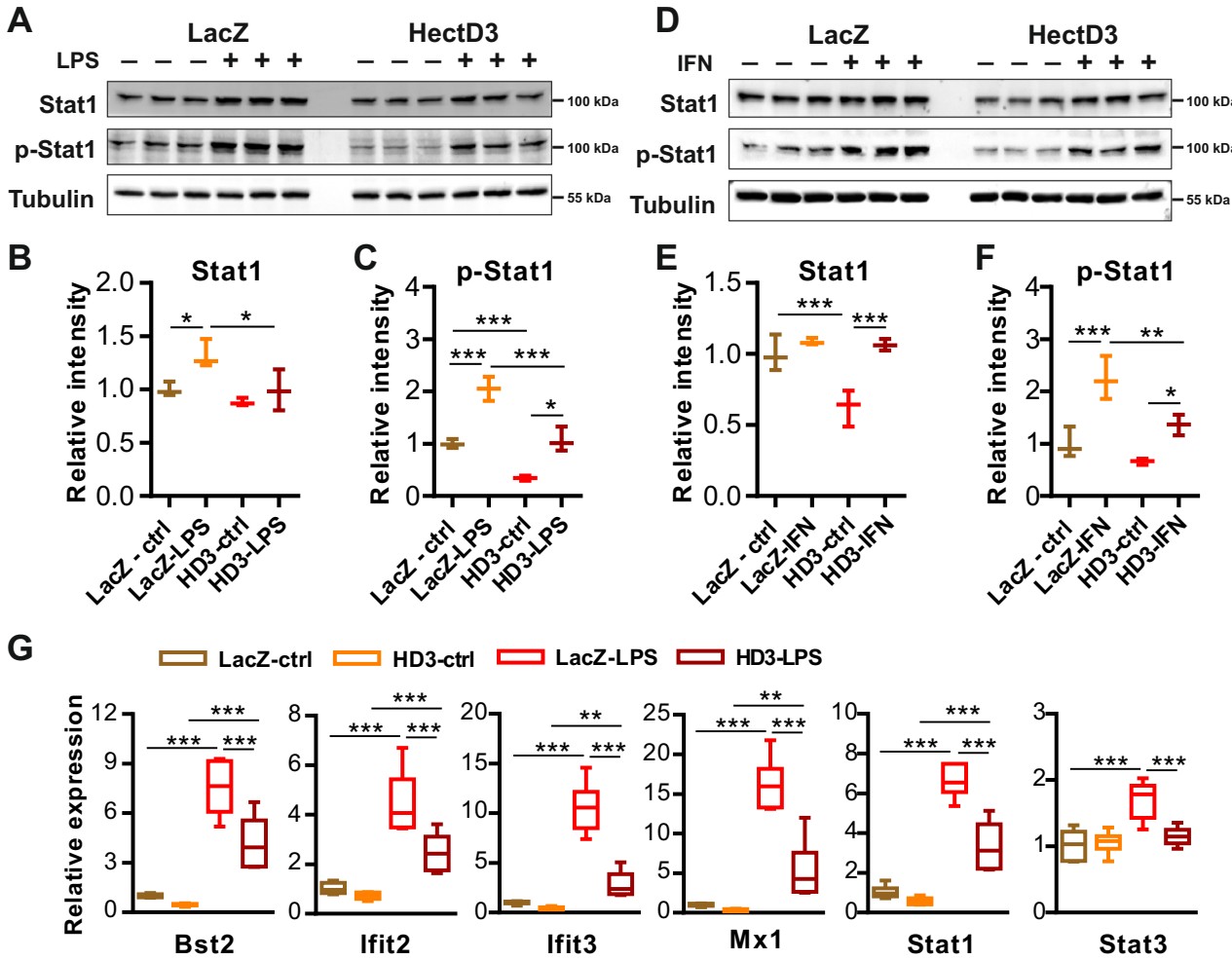

**Fig. 4 HectD3 inhibits LPS-/IFNγ-mediated activation of the inflammatory response in cardiomyocytes. a** Immunoblots displaying the expression of Stat1/p-Stat1 in control and HectD3 overexpressing NRVCMs in the absence or the presence of LPS. Densitometry analyses of **a** for Stat1/p-Stat1 are shown in **b** and **c** indicating that LPS significantly activated Stat1 by its phosphorylation, while HectD3 overexpression blunted this activation. ($n = 3$ each). **d** Immunoblots displaying the expression of Stat1/p-Stat1 in control and HectD3 overexpressing NRVCMs in the absence or the presence of IFNγ. Densitometry analyses of **b** for Stat1/p-Stat1 are shown in **e** and **f** indicating that IFNγ significantly activated Stat1 by its phosphorylation, while HectD3 overexpression blunted this activation ($n = 3$ each). **g** Transcript levels of several interferon responsive Stat1-signaling genes determined by quantitative real-time PCR revealing the same trend, i.e. upregulation upon LPS treatment in controls (LacZ) and reduction in upregulation upon HectD3 overexpression. ($n = 6$ each). Statistical calculations were carried out by two-way ANOVA. *$p < 0.05$, **$p < 0.01$, ***$p < 0.001$.

reduction in the number of resident cardiac macrophages (Fig. 6g, h). Finally, to determine whether HectD3 overexpression can directly alter the migratory capacity of macrophages through secretion of chemoattractant cytokines, we next performed trans-well migration assays using the RAW264.7 cell line. RAW264.7 cells were treated with the culture supernatant isolated from NRVCMs where HectD3 was either overexpressed or knocked-down (and respective controls) in the absence or the presence of LPS. As anticipated, macrophages demonstrated a robust increase in migration towards culture supernatant medium collected from control cardiomyocytes following LPS stimulation as compared to control non-treated cells (Fig. 6i, j). Conversely, migratory capacity was markedly abrogated by HectD3 overexpression in cardiomyocytes, while adenovirus-mediated HectD3 downregulation again promoted RAW264.7 cell migration (Fig. 6i, j).

Collectively, these findings suggest a role for HectD3 in reducing secretion of pro-inflammatory and chemoattractant factors via inhibition of Stat1, thereby attenuating the cardiac inflammatory response.

## Discussion

We recently reported SUMO2 as a potent activator of cardiac hypertrophy via activation of calcineurin-NFAT signaling[6]. Consistently, using transgenic mouse models overexpressing constitutively active SUMO2 in the heart, Kim et al. demonstrated the development of cardiomyopathy in four independent mouse lines[25]. The cardiac phenotype ranged from acute heart failure and early death to the progression of chronic cardiomyopathy with aging[25]. Importantly, both studies also reported a marked increase in SUMO2/3-mediated sumoylation in the hearts of human patients suffering from dilated, ischemic, or hypertrophic cardiomyopathy[6,25], pointing towards a broader pathogenic role of SUMO2 in the development of heart failure. We now report the ubiquitin ligase HectD3 as a novel cardiac-binding partner and upstream post-translational regulator of SUMO2 and SUMO2-mediated sumoylation. Moreover, our proteomics/transcriptomics data reveal Stat1 as a direct target of HectD3, as a result of which, HectD3 regulates inflammatory signaling in cardiomyocytes. These data thus provide a novel

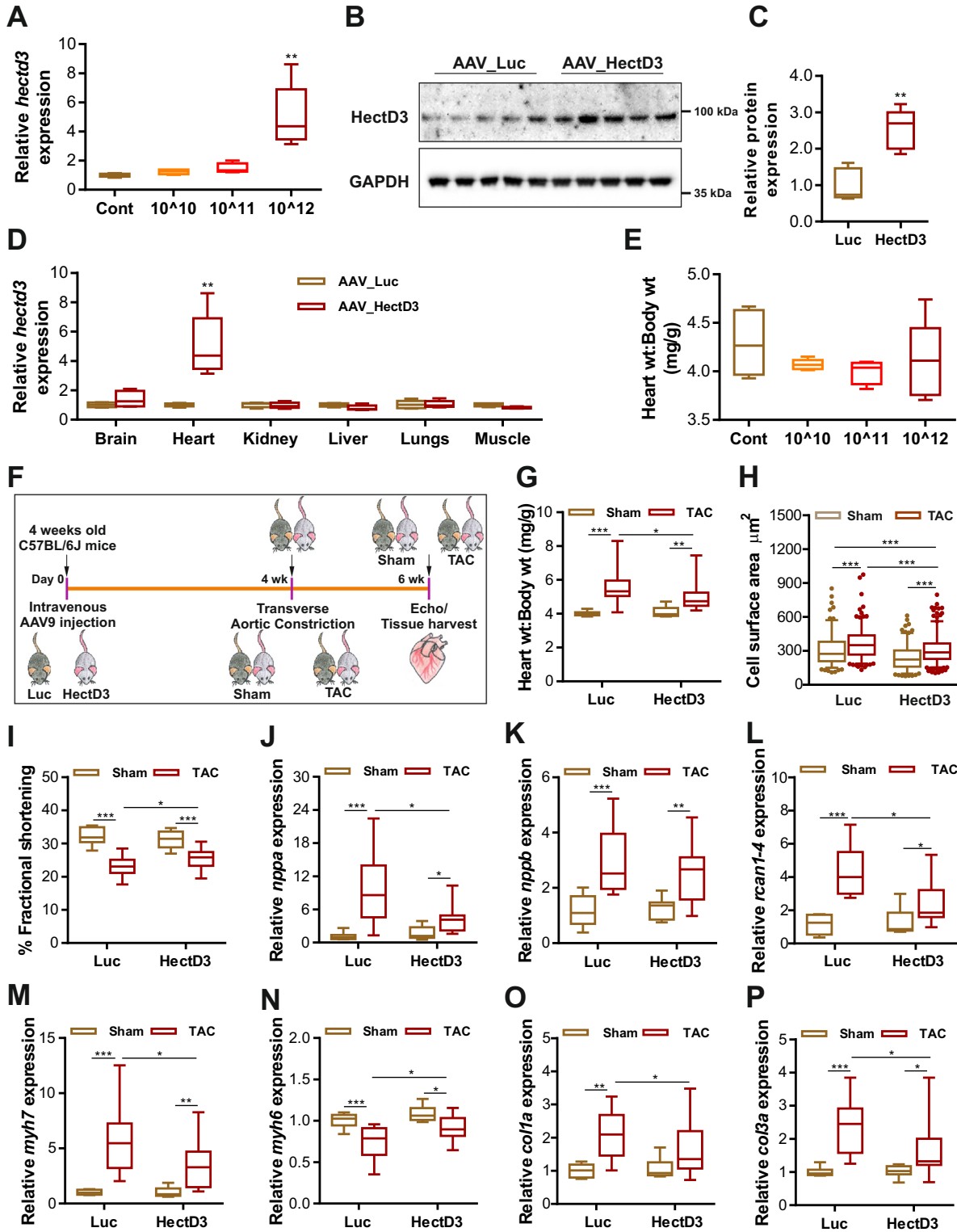

mechanism of dual regulation of cardiac hypertrophy and inflammation by HectD3 via dual regulation of SUMO2 and its sumoylation target Stat1.

HectD3 is known to interact with its target protein (s) through the APC10 domain to transfer the ubiquitin moiety from an E2-conjugating enzyme. As we also observed an interaction of HectD3 with SUMO2 through its APC10 domain, these data implied SUMO2 to be a possible direct target for HectD3 ubiquitination. We could confirm this hypothesis both in NRVCMs

and mouse hearts, where overexpression of HectD3 significantly accelerated polyubiquitination and degradation of SUMO2 with subsequent inhibition of downstream sumoylation. By virtue of post-translationally regulating SUMO2 and its target proteins, HectD3 attenuated the prohypertrophic effects of pressure overload due to TAC or AngII treatment. Mechanistically, we earlier reported that SUMO2 interacts with and tethers calcineurin to the nucleus, thereby accelerating the induction of calcineurin-NFAT signaling[6]. Consistently, we now also found that

**Fig. 5 HectD3 suppresses SUMO2-dependent activation of calcineurin-signaling and cardiac hypertrophy. a** Relative HectD3 expression in mouse hearts where HectD3-AAV was injected intravenously to find a suitable dose for further experiments. HectD3 was found to be significantly expressed only in mice injected with $1 \times 10^{12}$ times vector/mouse genome ($n = 5$ each). **b** Immunoblots further confirms the upregulation of HectD3 in mouse hearts harvested from mice injected with HectD3 ($1 \times 10^{12}$ times vector/mouse genome) compared to the AAV-luciferase (control), which is presented as a box plot in **c** ($n = 5$ each). **d** Relative HectD3 expression in various tissues obtained from mice injected with AAV-HectD3 demonstrates significant upregulation of HectD3 only in the heart compared to other tissues ($n = 5$ each). **e** Box plot indicating the heart weight (wt):body wt ratio in mice that were injected with different doses of AAV-HectD3. Data indicate no significant effect of HectD3 overexpression ($n = 5$ each). **f** Schematic outline of the experimental workplan for AAV-mediated overexpression of HectD3 in mice, including transverse aortic constriction (TAC) and downstream experiments. **g** Box plot indicating the heart weight (wt):body wt ratios in mice that underwent TAC or sham operations compared to the respective control group where HectD3 or luciferase (Luc) was overexpressed using equivalent AAV9 particles. Data indicates that TAC significantly increased the heart weight (wt):body wt ratio in control mice, whereas, AAV9-mediated overexpression of HectD3 reduced the ratio ($n = 8$ (AAV-Luc sham), 9 (AAV-HectD3 sham), 13 (AAV-Luc TAC), 14 (AAV-HectD3 TAC)). **h** Similarly, increased HectD3 expression reduced the cardiomyocyte cell surface area in mice that underwent TAC compared to the respective Luc mice group ($n > 150$ each). **i** Fractional shortening (%) as a measure of contractile function was also moderately improved in TAC operated mice when HectD3 was overexpressed. Expression of fetal genes determined by quantitative real-time PCR indicates upregulation of *nppa* **j**, *nppb* **k**, *rcan1.4* **l**, β-myosin heavy chain (*myh7*) **m**, and downregulation of α-myosin heavy chain (*myh6*) **n** after TAC; HectD3 overexpression however reduced this up-/down-regulation ($n = 8$ (AAV-Luc sham), 9 (AAV-HectD3 sham), 13 (AAV-Luc TAC), 14 (AAV-HectD3 TAC)). Similarly, TAC upregulated fibrosis markers Collagen 1a (*col1a*) **o**, and 3a (*col3a*) **p** which was again attenuated by HectD3 overexpression ($n = 8$ (AAV-Luc sham), 9 (AAV-HectD3 sham), 13 (AAV-Luc TAC), 14 (AAV-HectD3 TAC)). Statistical calculations were carried out by two-tailed Student's t-test or two-way ANOVA. *$p < 0.05$, **$p < 0.01$, ***$p < 0.001$.

overexpression of HectD3 substantially reduced nuclear localization of active calcineurin, likely due to reduced SUMO2–calcineurin interaction secondary to lower levels of SUMO2 in the presence of upregulated HectD3. While polysumoylation of the target proteins by SUMO2/3 has been reported earlier to facilitate ubiquitin-mediated degradation, this is, to the best of our knowledge, the first report of a direct regulation of SUMO2 and overall sumoylation via ubiquitination.

Accumulating evidence also suggests that pressure overload and AngII infusion drive the generation of a pro-inflammatory cellular environment, characterized by intense secretion of cytokines, chemokines and factors that stimulate migration of monocytes and their differentiation into macrophages in the cardiac tissue[28–30]. In turn, the expansion of immune cells promotes pathological tissue remodeling[31] and induces cardiomyocyte apoptosis and ultimately cardiac decompensation. In line with previous studies, we could detect a marked increase in F4/80 positive cells in response to TAC and AngII application. Notably, our findings suggest that HectD3-overexpression in cardiomyocytes could significantly attenuate the accumulation of macrophages in the two investigated in vivo models of cardiac hypertrophy and heart failure. Furthermore, we could demonstrate a simultaneous decrease in the mRNA levels of pro-inflammatory and chemoattractant markers which may account for the observed phenotype. Interestingly, inhibition of IL-6 has been shown to attenuate AngII-induced cardiac hypertrophy and fibrosis[32], and cardiac-deficiency of the NF-κB subunit, p65, led to reduced cardiac hypertrophy, and preserved contractile function after TAC[33]. While clinical trials for anti-inflammatory therapies in heart failure such as the use of TNFα antagonists yielded mixed results (reviewed in refs. [34,35]), there is renewed interest in such an approach as e.g. a sub-analysis of the CANTOS trial revealed that interleukin 1β-blockade improved ejection fraction and heart failure symptoms[36].

SUMO2/3 has recently been described to prevent both the canonical and non-canonical interferon response in inflammatory cells[18,19]. In particular relevance to our current findings, several studies have shown that SUMO-conjugation of Stat1 by SUMO1 or SUMO2/3 inhibits its activity due to reduced phosphorylation and reduced DNA-binding efficiency on Stat1-responsive gene promoters, which in turn impedes the induction of the inflammatory response[19,37,38]. In contrast to these findings in other cell types, we observed a positive correlation between SUMO2 levels and Stat1 activation in cardiomyocytes, where overexpression of SUMO2 significantly increased phosphorylation of Stat1. HectD3 suppressed this activation not only by reducing SUMO2

levels but also by directly acting as an E3 ubiquitin ligase of Stat1. Signal transducers and activators of transcription (Stat) is a family of interferon responsive transcription factors, which is known to mediate cytokine-driven intracellular signaling. Out of seven known family members (Stat1–Stat7), Stat1 and Stat3 have been extensively studied in cardiac pathophysiology[39]. Interestingly, Stat1 and Stat3 have been suggested to play antagonistic roles, where Stat1 worsens, while Stat3 protects the heart from ischemic cardiomyopathy by either activating or inhibiting inflammation, respectively[39–41]. Stat1 has been shown to be one of the major players orchestrating the immune response activation contributing to the pathogenesis of various cardiovascular diseases, such as atherosclerosis[42,43], myocardial infarction[39], heart failure[44], and cardiac allograft vasculopathy[45]. Increased Stat1-dependent gene expression is associated with abnormal levels of inflammatory markers[46], cardiomyocyte apoptosis[47], and enhanced extracellular matrix deposition[48]. In our study, we found that HectD3 mediated a reduction in Stat1 activation without impairing Stat3 signaling thereby protecting the myocardium against pathological hypertrophy induced by TAC and AngII infusion. In line with our data, it was previously shown that cardiomyocyte-specific Stat1 deficiency is cardioprotective in a mouse model of ischemia-reperfusion injury[40]; nevertheless, few studies so far have focused on the contribution of Stat1 in pathological myocardial hypertrophy and subsequent heart failure.

In conclusion, we here describe a novel cardioprotective mechanism involving the ubiquitin ligase HectD3, which exerts anti-hypertrophic and anti-inflammatory effects likely via dual regulation of SUMO2 and its sumoylation target Stat1 (Fig. 7). Our data lend support to the notion that hypertrophy and inflammation are more intertwined than previously appreciated. We thus propose HectD3 as a "dual pathway" inhibitor for the prevention and/or therapy of cardiac hypertrophy and heart failure, e.g. utilizing AAV-mediated gene transfer.

## Methods

All animal experiments were approved by the Ministry of Energy Transition, Agriculture, Environment, Nature and Digitalization (MELUND) of the state of Schleswig-Holstein and were carried out stringently following international and institutional ethical guidelines. The use of human tissue samples conforms to the declaration of Helsinki and was approved by the ethical committee of the medical school of the Georg-August-University, Göttingen. Written informed consent was received from all participants prior to inclusion.

**Isolation and culture of neonatal rat cardiomyocytes**. NRVCMs were isolated from 1 to 2 days old Wistar rats (Charles River, Germany) as previously

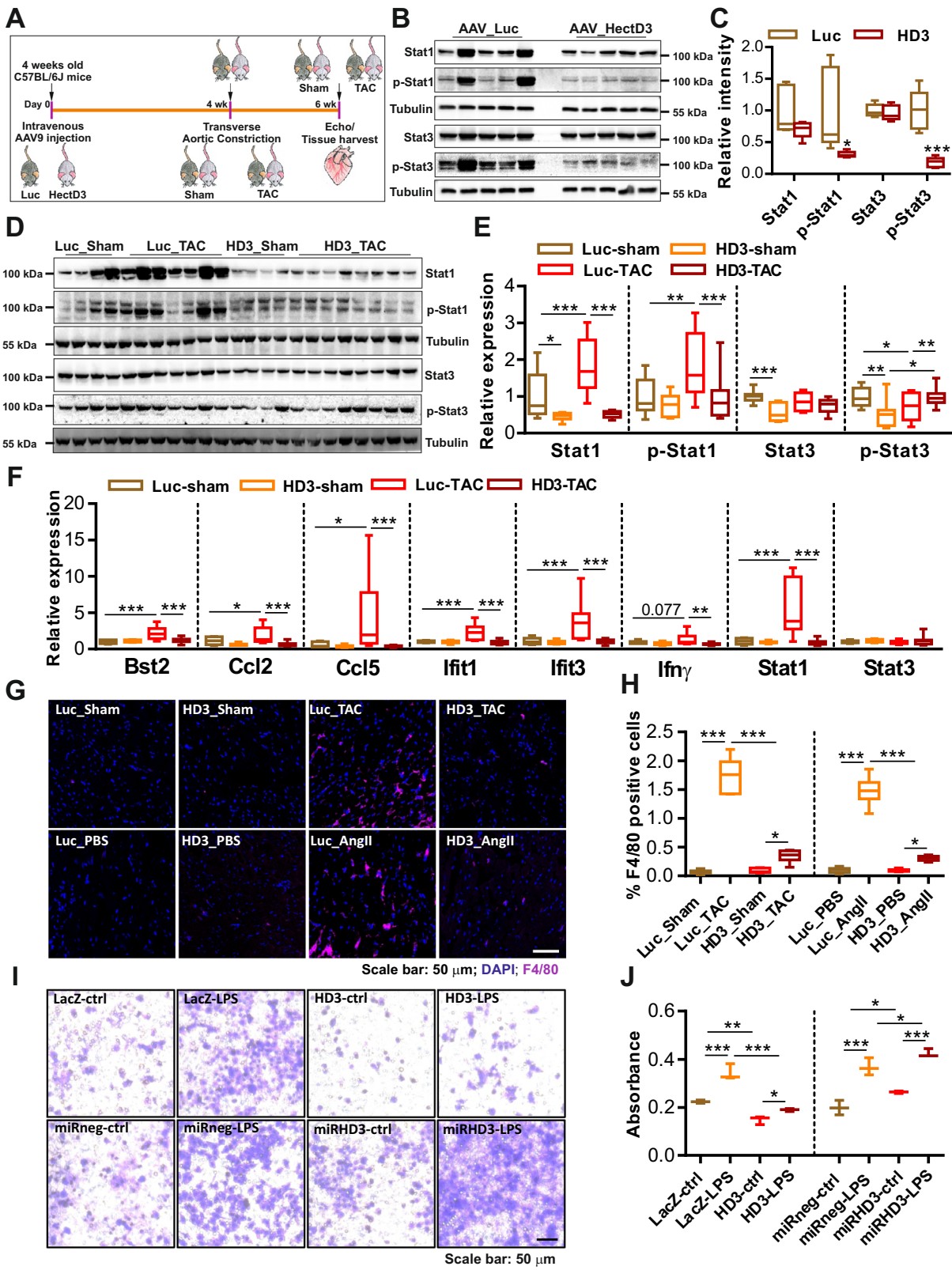

described[49,50]. Briefly, neonatal rats were decapitated to collect hearts in ADS buffer (120 mmol/l NaCl, 20 mmol/l HEPES, 8 mmol/l $NaH_2PO_4$, 6 mmol/l glucose, 5 mmol/l KCl, and 0.8 mmol/l $MgSO_4$ (pH 7.4)). All the steps henceforth are performed in sterile conditions. Ventricles were chopped and digested in 4–5 steps with collagenase type II (0.5 mg/ml, Worthington Biochemical Corporation, Germany) and pancreatin (0.6 mg/ml, Sigma-Aldrich, Germany) in ADS buffer at 37 °C. Cardiomyocytes and fibroblasts were separated using a Percoll gradient

centrifugation step. Isolated cardiomyocytes were then cultured in DMEM supplemented with 10% (v/v) FCS, 100 U/ml penicillin and 100 µg/ml streptomycin (Life Technologies), and 2 mmol/L L-glutamine (Life Technologies) for 24 h. Treatments of overexpression/knockdown of desired proteins are achieved by adenovirus-mediated transduction for 72 h in serum-depleted media. NRVCMs were also treated with either IFNγ or LPS in some experiments as 1 µg/ml or 100 ng/ml for 24 or 12 h, respectively.

**Fig. 6 HectD3 attenuates Stat1-mediated inflammatory response in vivo. a** Schematic outline of the experimental work-plan for AAV-mediated overexpression of HectD3 in mice, including TAC and downstream experiments. **b** Immunoblots indicating proteins levels of Stat1/p-Stat1 and Stat3/p-Stat3 in mouse hearts where HectD3 was overexpressed via AAV-mediated gene transfer. AAV-luciferase injected mice were used as a control group. **c** Densitometry analysis of images shown in **b** is presented as a box plot that shows a strong reduction in the phosphorylation of both Stat1 and Stat3 by HectD3 overexpression ($n = 5$ each). **d** Immunoblots indicating the protein levels of Stat1/p-Stat1 and Stat3/p-Stat3 after TAC or sham surgery in mouse hearts where HectD3 was overexpressed via AAV-mediated gene transfer. AAV-luciferase injected mice were used as control group. **e** Densitometry analysis of images shown in **d** are presented as box plots for Stat1, p-Stat1, Stat3, and p-Stat3 showing that the phosphorylation of Stat1 was increased, whereas phosphorylation of Stat3 was reduced after TAC in control mice. HectD3 overexpression strongly reduced the phosphorylation of Stat1, yet maintained the phosphorylation levels of Stat3 comparable to control sham mice after TAC. **f** Transcript levels of several inflammatory markers and downstream targets (*Bst2, Ccl2, Ccl5, Ifit1, Ifit3, IFNγ, Stat1,* and *Stat3*) of interferon-signaling and Stat1-signaling are detected by quantitative real-time PCR in mouse hearts after TAC in mouse hearts where HectD3 was overexpressed by AAV-mediated gene transfer. **g** Representative immunofluorescence microscopy images of cardiac tissue sections of mouse hearts obtained from luciferase (control) or HectD3 injected mice that underwent TAC operations or Angiotensin-II (AngII) infusion (with respective sham or PBS controls). Tissue sections were stained with F4/80 as a marker of macrophage and counterstained with DAPI for nuclear staining. HectD3 overexpression reduced the infiltration of inflammatory macrophages to the mouse hearts after TAC compared to the respective control mice. **h** Immunofluorescence intensity analysis of F4/80 positive cells are presented in the form of a box plot indicating a dramatic increase in the macrophages in TAC operated or AngII-treated control mice. HectD3 overexpression however significantly reduced macrophage numbers ($n = 6$ each). **i** Representative images of RAW 264.7 cells after chemotaxis assay in modified Boyden chambers containing polycarbonate membranes. **j** Optical density measurements are presented in the form of a box plot showing accelerated migration of RAW 264.7 cells when treated with LPS. However, cells treated with culture supernatant from NRVCMs overexpressing HectD3 reduced this migration. Contrasting results were obtained with HectD3 knockdown condition ($n = 3$ each). Statistical calculations were carried out by two-way ANOVA. *$p < 0.05$, **$p < 0.01$, ***$p < 0.001$.

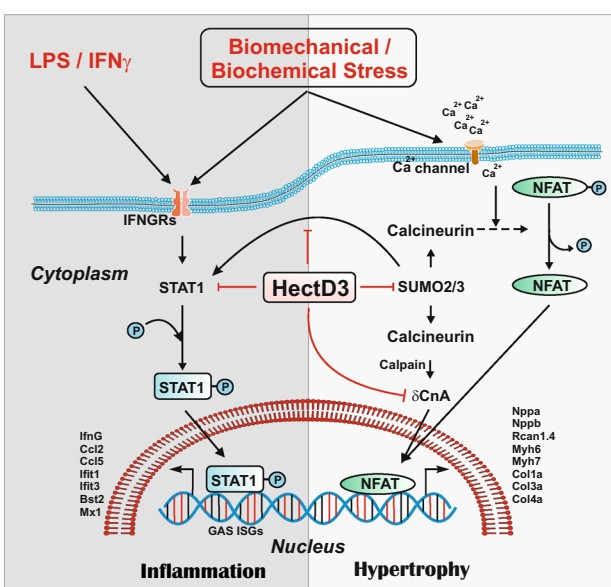

**Fig. 7 Schematic representation of HectD3-mediated dual inhibitory regulation of cardiac hypertrophy and inflammation.** Right half panel indicate that HectD3 targets SUMO2 for UPS-mediated degradation, subsequently attenuating nuclear localization of constitutively active calcineurin A (δCnA) and activation of CnA-NFAT hypertrophic signaling in vitro and in vivo. On the other hand, left half panel demonstrates that HectD3 interacts with and targets interferon response factor Stat1 thereby downregulating pro-inflammatory Stat1-target genes, consequently resulting in reduced secretion of chemoattractants, and infiltration of pro-inflammatory macrophages to the heart.

**RAW 264.7 culture**. RAW264.7 mouse macrophage cell line was purchased from American Type Culture Collection (ATCC, Manassas, VA). Cells were maintained in DMEM medium, supplemented with 10% FBS, 1% glutamine, and 1% penicillin/streptomycin.

**Macrophage migration assay**. Chemotaxis assay was performed as previously described[51]. In brief, modified Boyden chambers containing polycabonate membranes with a pore size of 8 μm (Merck, Darmstadt, Germany) were placed into the lower chamber containing 1 mL of NRVCM-derived conditional medium. The supernatants were collected 12 h after treatment with 100 nM LPS. Serum-starved

RAW 264.7 macrophages were added on top of each chamber and incubated at 37 °C for 24 h to allow cell migration. Afterwards, non-migratory cells on the upper surface of the membrane were removed and adherent cells on the bottom were stained with 2.3% crystal blue solution (Merck, Darmstadt, Germany) for 10 min at room temperature. Migratory cells were visualized using an inverted microscope. Next, transwell membranes were cut and incubated with 200 μL acetic acid to induce the solubilization of the dye. Optical density was measured spectro-photometrically at a wavelength of 560 nm (Tecan, Männedorf, Switzerland). All source data underlying the graphs presented in the main figures using migration assay are made available as Supplementary Data 1.

**AAV9-mediated gene-transfer**. The experimental strategy for AAV-mediated gene-transfer is diagrammatically presented in Fig. 5f and Supplementary Fig. 5D. Briefly, single strand AAV vector genomes harboring either the HectD3 or luci-ferase cDNA under transcriptional control of the human troponin T promoter (Tnt-4) were used for generation of AAV9 vectors as described before[24,52]. Respective AAVs (up to $10^{12}$ vector genomes per animal in a total of 100 μL PBS) were injected through the tail-veins of 4 week old mice C57BL/6 mice purchased from Charles River Laboratories. Mice underwent TAC or sham operations (detailed below) or AngII infusion 4 weeks post injections and kept for 2 more weeks under controlled 12/12 h dark/light cycle. Animals were assessed for cardiac function by echocardiography and sacrificed to harvest organs for downstream applications.

**TAC and echocardiography**. TAC/sham surgeries were carried out in 8 weeks old C57BL/6 mice (4 weeks post AAV injections, operator was blinded to AAV injection sub-groups as described[53]. Briefly, mice were analgized by Buprenor-phine (0.1 mg/kg s.c.) injection and kept on a heating pad to maintain the body temperature at 37 °C. Mice were anesthetized with 2.5 vol% isoflurane in 0.2 L/min O₂ and orally intubated with 20-gauge tube and ventilated (Harvard Apparatus) at 120 breaths per minute (0.2 mL tidal volume). A small incision in second costal space was made, to expose aortic arch. A ligation with suture (Prolene 6-0) and against needle 27 G (Ø 0.42 mm) was placed around the aorta between the bra-chiocephalic and left carotid artery. By immediately restoring blood flow, the needle was removed. With suture (Prolene 4-0), the chest was closed and the pneumothorax evacuated. For control group (sham), mice underwent the same surgery without the ligation. Mice were examined 2 weeks post-surgery by Echo-cardiography on VisualSonics Vevo 1100 and MS400 cardiovascular probe (18–38 MHz). To obtain following parameters of cardiac function left ventricle (LV) interventricular septal thickness (IVS), LV internal dimensions (LVID), and pos-terior wall (PW) thicknesses at diastole and systole (IVSd, LVIDd, PWd and IVSs, LVIDs, PWs) were measured from M-mode short-axis images at the level of the papillary muscles. Left ventricular fractional shortening (FS) was assessed by B-mode long axis images.

**Human HCM patient heart samples**. Left ventricular heart tissue was collected from explanted hearts of HCM patients undergoing heart transplantation during surgical procedures and placed immediately in pre-cooled cardioplegic solution (110 mM NaCl, 16 mM KCl, 16 mM MgCl₂, 16 mM NaHCO₃, 1.2 mM CaCl₂, and 11 mM glucose). Non-failing hearts explanted from accidental death patients were used as controls. Myocardial samples for immunoblotting/qPCR analysis were

snap-frozen in liquid nitrogen and stored at −80 °C immediately after excision. All procedures involving human samples were approved and performed in strict compliance with the ethical committee of the medical school of the Georg-August-University, Göttingen.

**Cloning SUMO2, HectD3, and Ubiquitin**. Full length coding regions of SUMO2, HectD3, and Ubiquitin were cloned in Gateway cloning entry vector pDONR221 and subsequently recombined into Gateway compatible destination vector pcDNA3.1 (to be used for expression in HEK293-A cells) or into pAd/CMV/V5 (to generate adenovirus for overexpression in NRVCMs) following manufacturer's instructions (Life Technologies). Synthetic microRNAs specifically targeting HectD3 (miR-HectD3) was generated using BLOCK-iT Pol II miR RNAi Expression system to knockdown respective genes (Life Technologies, Germany). Finally, adenoviruses encoding SUMO2, HectD3, Ubiquitin, and synthetic microRNAs were produced using the ViraPower™ Adenoviral Kit according to the manufacturer's instructions (Life Technologies, Germany). A β-Galactosidase-V5 encoding adenovirus (Ad-LacZ, Life Technologies) and an unspecific synthetic microRNA (miR-Neg) served as control viruses for overexpression and knockdown experiments, respectively. All the cloning primers used in this study are listed in Supplementary Data 2.

**Generation of AAV9-vectors**. The full-length coding sequence of HectD3 was cloned into a single-stranded AAV vector backbone (pSSV9-CMV-MLC1500-luc). AAV9 vectors were produced in HEK293A cells by co-transfection of helper plasmid pDP9rs and either pSSV9-CMV-MLC1500-HectD3, or pSSV9-CMV-MLC1500-luciferase (as a control), and purified using iodixanol step gradient ultracentrifugation as reported before[52].

**Antibodies**. Antibodies used in this study are listed below where respective dilutions are mentioned in the parentheses:

α-actinin, mouse monoclonal, Sigma-Aldrich (1:400); SUMO2/3, rabbit monoclonal, Cell signaling (1:1000); SUMO2 + 3, mouse monoclonal, Abcam (1:1000); Calcineurin A, mouse monoclonal, BD Bioscience (1:250); GAPDH, mouse monoclonal, Sigma-Aldrich (1:20,000); Histone H3, rabbit polyclonal, Cell-signaling (1:1000); α-Tubulin, mouse monoclonal, Sigma-Aldrich (1:8000); Ubiquitin, mouse monoclonal, Millipore (1:1000); STAT1, rabbit monoclonal, Cell signaling (1:1000); p-STAT1, rabbit monoclonal, Cell signaling (1:1000); STAT3, rabbit monoclonal, Cell signaling (1:1000); p-STAT3, rabbit monoclonal, Cell signaling (1:1000); F4/80, rat monoclonal, Dianova (1:500); HA, mouse monoclonal, Sigma-Aldrich (1:20,000); HectD3, rabbit polyclonal, Mybiosource (1:1000); V5, mouse monoclonal, Biozol (1:1000).

**Protein preparation and immunoblotting**. NRVCMs were lysed 72 h post viral transduction by three freeze–thaw cycles in RIPA buffer (20 mM Tris, 10 mM DTT, 500 mM sodium chloride, 1% NP40, 12,5% glycerol) supplemented with phosphatase and protease inhibitor cocktails (Roche, Germany). The cell lysate was then centrifuged at $12,000 \times g$ for 20 min. to remove cell debris, and the supernatant was used for protein concentration measurement by DC-assay (Bio-Rad Laboratories). Proteins prepared as above were resolved on 10% SDS–PAGE, or commercially available 4–12% gradient gels (Life Technologies), and transferred to a polyvinylidenefluoride membrane, blocked for 2 h in 5% dry-milk prepared in 0.1% TBST at room temperature (RT), followed by the incubation with primary antibodies overnight at 4 °C, 4× washes with 0.1% TBST and final incubation with a suitable HRP-coupled secondary antibody (1:10,000) (Santa Cruz, Germany). Protein signals were developed using ECL-select chemiluminescence kit (GE Healthcare) and visualized on Fluorchem Q imaging system (Biozym). Quantitative densitometry was performed with the help of ImageJ/Fiji version 1.46. All the uncropped immunoblot images have been included as Supplementary Fig. 7. All source data underlying the graphs presented in the main figures using immunoblot densitometry data are made available as Supplementary Data 1.

**Immunoprecipitation in protein lysate from mouse heart**. Total protein extracted from mouse heart was used for immunoprecipitation of endogenous binding partners of SUMO2. Approximately 4 μg of anti-SUMO2/3 antibody (mouse monoclonal, Abcam) was incubated with 1 mg total protein in 1 mL RIPA buffer for 6 h at 4 °C in 1.5 mL microcentrifuge tubes on a rotating mixer. Dynabeads (50 μl, Life Technologies, Germany) equilibrated with RIPA buffer were pipetted in the protein–antibody mix and incubated overnight at 4 °C on rotating mixer. Protein lysate was carefully removed after placing microcentrifuge tubes on a magnetic stand. Beads were then washed six times with the lysis buffer. Precipitated proteins were eluted with 50 μl of 2× Laemmli buffer by boiling the mix at 95 °C for 5 min. The eluted protein was determined by immunoblotting.

**Immunoprecipitation in protein lysate from HEK293A cells**. HEK293A cells were maintained in DMEM containing 4% FCS, 2 mM L-glutamine, and penicillin/streptomycin. STAT1 (Gene ID: 25124) cDNA clone (pCS6 (BC062079)-TCR1305-GVO-TRI) was purchased from Biocat GmbH (Germany); HectD3 and V5-SUMO2/3 overexpressing clones were cloned in pDEST40; while prK5-HA-

Ubiquitin-WT was a kind gift from Dr. Ted Dawson (Addgene plasmids). For establishing interaction between STAT1 or SUMO2/3 and HA-Ubiquitin in presence of HectD3, HEK293A cells ($2.5 \times 10^6$) were co-transfected with each of STAT1 or SUMO2/3 (15 μg) and HA-Ubiquitin (WT or mutants, 10 μg each) expression plasmids with or without HectD3 (15 μg) using Lipofectamine 2000 (Life Technologies, Inc.). Empty vector pcDNA3.1 was used as a negative control. 48 h after transfection with an intermittent media change at 24 h, cells were washed with PBS, pelleted down, and resuspended in buffer A (50 mM Tris, 150 mM NaCl, 1% Nonidet P-40, 0.5% sodium deoxycholate, and 0.2% SDS) supplemented with phosphatase and protease inhibitor mixtures (Complete; Roche Applied Science). Cells were lysed by three successive freeze–thaw cycles with one sonication step for 10 s (Bandelin sonoplus, at 40% power), followed by centrifugation at $18,000 \times g$ at 4 °C for 20 min. The supernatant containing cellular proteins was used for immunoprecipitation using Anti-HA-tag mAb-Magnetic Beads (MBL international) for Stat1-related IP and Anti-V5-tag mAb-Magnetic Beads (MBL international) for SUMO2/3-related IP following the manufacturer's guidelines. In brief, 500 μg of protein in a total volume of 1 ml of lysis buffer was applied to 50 μl of equilibrated beads, and proteins were then allowed to bind to the HA/V5-tagged antibody on the beads for ~14 h at 4 °C atop a rotor. Protein lysates were removed after centrifugation at $8000 \times g$, and magnetic beads were washed three times with lysis buffer with help of a magnetic stand. Precipitated proteins from the beads were then eluted with 50 μl of Laemmli sample buffer. An amount of 10 μl from this eluted protein was immunoblotted with SDS–PAGE followed by transfer to nitrocellulose membranes and developed against respective antibodies to confirm interactions between STAT1 or SUMO2/3 and HA-Ubiquitin proteins in presence of HectD3.

**In vitro ubiquitination assay**. HEK293A cells were transfected with plasmids encoding Stat1 together with either HectD3 or ubiquitin and HectD3. Protein lysates (50 μg) from respective conditions were incubated in 50 μl ubiquitination buffer (RIPA buffer supplemented with phosphatase and protease inhibitor cocktails (Roche, Germany), MG-132 [10 μM, Sigma], ubiquitin-aldehyde [0.5 μM, Enzo Lifesciences], DTT [1 mM], MgCl₂ [2 μM] and ATP [5 mM]) for 30 min at 37 °C. Samples were resolved by SDS–PAGE and immunoblotted against Stat1.

**Immunofluorescence microscopy**. Co-localization of SUMO2 and HectD3 or cell surface area was determined by immunofluorescence microscopy. Briefly, NRVCMs were seeded on collagen-coated glass-coverslips in 12× well cell-culture plates and grown in 10% serum containing DMEM-medium with glucose, L-Gln, and Pen/Strep for 24 h. NRVCMs were then either transduced with viruses for 72 h or directly fixed with 4% paraformaldehyde for 10 min at RT, permeabilized, washed, and blocked with 0.1% Triton X-100 in PBS with 2.5% BSA at RT for 1 h. Cells were then incubated with α-actinin for cell surface area measurement or with SUMO2 + HectD3 primary antibodies for co-immunostaining under humidified conditions. Respective secondary antibodies conjugated with AlexaFluor488 and/or AlexaFluor546 (Invitrogen, Germany) were incubated for 1 h, at RT. Images were captured with a Keyence BZ-9000 fluorescence microscope using ×20 CFI Plan Apo λ lens (NA of 0.75) with the in-built CCD-camera at RT (Keyence, Japan). Images were processed and analyzed by BZ-II Analyzer (Keyence, Japan). The cell surface area was measured as described in detail before[54]. All source data underlying the graphs presented in the main figures using immuno-fluorescence microscopy data are made available as Supplementary Data 1.

**Proteomics analysis by LC–MS/MS**. NRVCMs were transduced with adenoviruses expressing either HectD3 or LacZ and incubated for 72 h. Cell pellets were then lysed in 100 μl buffer containing 6 M guanidinium hydrochloride, 100 mM HEPES pH 7.5, and protease inhibitors. Reduction and alkylation of cysteines were carried out by the addition of 10 mM DTT and 55 mM iodoacetamide, followed by tryptic digestion at 37 °C overnight. Peptides were subsequently extracted using C18 SepPak columns, lyophilized and resuspended in 100 μl of 50 mM TEAB and subjected to TMT-labeling. After a second solid phase, extraction the samples were fractionated into 10 fractions using high pH reversed-phase chromatography to increase proteome coverage[55].

LC–MS/MS analysis was performed using a Dionex U3000 nanoUHPLC coupled to a Q Exactive Plus mass spectrometer (Thermo Scientific) using following parameters: 6 μl sample was injected and loaded on a trap column (Acclaim Pepmap 100 C18, 10 mm × 300 μm, 3 μm, 100 Å, Dionex) and washed for 3 min with 2% ACN/0.05% TFA at a flow-rate of 30 μL/min. Separation was performed using an Acclaim PepMap 100 C18 analytical column (50 cm × 75 μm, 2 μm, 100 Å, Dionex) with a flow-rate of 300 nL/min and following eluents: A (0.05% FA) and B (80% ACN/0.04% FA); linear gradient 5–40% B in 180 min, 50–90% B in 5 min, 90% B for 10 min, 90–5% B in 1 min, and equilibrating at 5% B for 11 min. Ionization was performed with 1.5 kV spray voltage applied on a non-coated PicoTip emitter (10 μm tip size, New Objective, Woburn, MA) with the source temperature set to 250 °C. MS data were acquired from 5 to 200 min with MS full scans between 300 and 1800 $m/z$ at a resolution of 70,000 at $m/z$ 200. The 15 most intense precursors with charge states ≥2+ were subjected to fragmentation with HCD with NCE of 27%; isolation width of 3 $m/z$; resolution, 17,500 at $m/z$ 200. Dynamic exclusion for 30 s was applied with a precursor mass tolerance of 10 ppm.

Lock mass correction was performed based on the polysiloxane contaminant signal of 445.120025 m/z. Additional wash runs were performed between samples from gel bands to reduce carry over while cytochrome C was used to monitor mass accuracy and LC quality control.

The acquired MS/MS-spectra were searched with the SequestHT algorithm against the entire reviewed protein database of *Rattus norvegicus* (in total 29,969 sequences, Proteome ID:UP000002494). Static modifications applied for searches were carbamidomethylation on cysteine residues, TMT modification of peptide N-terminus, and lysines while oxidation on methionine residues was set as dynamic modifications. Spectra were searched with semi-enzyme specificity. A MS mass tolerance of 10 ppm and a MS/MS tolerance of 0.02 Da was used. Peptide grouping was applied to group Peptides Spectrum Matches (PSMs, a MS/MS-spectrum assigned to a peptide sequence from the protein database) with the same modification under the same peptide. The protein group identifications were further filtered based on the following criteria: Proteins must be identified with at least two peptides with a FDR confidence ≤0.05 (medium).

TMT reporter ions were used for relative quantification between control and biological treatment in the three biological replicates. Only protein IDs with reporter signals were considered during the analysis. All source data underlying the graphs presented in the main figures using mass spectrometry data are made available as Supplementary Data 1.

**Reporter gene assays**. All the reporter gene assays are performed in NRVCMs. Cells were transduced with combinations of different viruses expressing SUMO2 (50 multiplicity of infection (moi)), HectD3 (50 moi), ΔCnA (50 moi), NFAT-reporter-luc (20 moi) carrying a firefly luciferase, AdRen-luc carrying renilla luciferase (5 moi, for normalization of the measurements), and LacZ as control or a filler virus to maintain equal virus load. For knockdown experiments, miR-HectD3 or miR-Neg adenoviruses were used. Luciferase activity was measured using a dual luciferase reporter assay kit following the manufacturer's instructions (Promega). Chemiluminescence was detected on infinite m200 PRO system (Tecan). All source data underlying the graphs presented in the main figures using luciferase reporter assays are made available as Supplementary Data 1.

**RNA isolation and quantitative real-time PCR**. Total RNA was isolated from NRVCMs 72 h post viral transduction using QIAzol lysis reagent (Qiagen) following the manufacturer's instructions. One microgram of DNA-free total RNA was transcribed using Superscript III first-strand cDNA synthesis kit (Life Technologies). Quantitative real-time PCR (qRT-PCR) was performed using EXPRESS SYBR GreenER Reagent (Life Technologies) in CFX96 real-time Cycler (Bio-Rad Laboratories). PCR conditions used were: 3 min at 95 °C, followed by 40 cycles of (15 s at 95 °C for denaturation, and 45 s at 60 °C, for annealing and extension, data was collected at this step). Rpl32 was used as an internal normalization control using delta-delta-Ct algorithm[56]. All the primers used in this study for qRT-PCR are listed in Supplementary Table 1.

**RNA-seq analysis**. Total RNA from each sample was quantified using a Nano-Drop ND-1000 instrument. Approximately 2 μg total RNA was used to prepare the sequencing library in the following steps: 1. Total RNA was enriched by oligo (dT) magnetic beads (rRNA removed); 2. RNA-seq library preparation using KAPA stranded RNA-Seq Library Prep Kit (Illumina), which incorporates dUTP into the second cDNA strand and renders the RNA-seq library strand-specific. The completed libraries were qualified with Agilent 2100 Bioanalyzer and quantified by absolute quantification qPCR method. To sequence the libraries on the Illumina HiSeq 4000 instrument, the barcoded libraries were mixed, denatured to single-stranded DNA in NaOH, captured on Illumina flow cell, amplified in situ, and subsequently sequenced for 150 cycles for both ends on Illumina HiSeq instrument.

Image analysis and base calling were performed using Solexa pipeline v1.8 (Off-Line Base Caller software, v1.8). Sequence quality was examined using the FastQC software (http://www.bioinformatics.babraham.ac.uk/projects/fastqc/). The trimmed reads (trimmed 5′, 3′-adapter bases using cutadapt[57]) were aligned to the reference genome using Hisat2 software[58]. The transcript abundances for each sample was estimated with StringTie[59], and the FPKM[60] value for gene and transcript level were calculated with R package Ballgown[61]. The differentially expressed genes and transcripts were filtered using R package Ballgown. The novel genes and transcripts were predicted from assembled results by comparing to the reference annotation using StringTie and Ballgown, then used CPAT[62] to assess the coding potential of those sequences. Then use rMATS to detecting alternative splicing events and plots[63]. Principle component analysis (PCA) and correlation analysis were based on gene expression level, hierarchical clustering, gene ontology, pathway analysis, gene ontology, pathway analysis, scatter plots, and volcano plots were performed with the differentially expressed genes in R, Python, or shell environment for statistical computing and graphics. All source data underlying the graphs presented in the main figures using RNA-seq data are made available as Supplementary Data 1.

**Statistics and reproducibility**. All the results are presented as box and whiskers plots with minimum to maximum range (showing all points), unless specified otherwise. Statistical significance was determined using two-tailed Student's t-test,

two-way analysis of variance (ANOVA, followed by Student–Newman–Keuls post-hoc test), or non-parametric Kruskal-Wallis test (with Dunn's multiple comparison test) where appropriate, details of which are also included in respective figure legends. *P*-values ≤ 0.05 were considered statistically significant.

**Reporting summary**. Further information on research design is available in the Nature Research Reporting Summary linked to this article.

## Data availability

The RNA-sequencing data was deposited to the gene expression omnibus (GEO) database (https://www.ncbi.nlm.nih.gov/geo/) and is available under the accession number GSE155768. The mass spectrometry proteomics data have been deposited to the ProteomeXchange Consortium via the PRIDE[64] partner repository with the dataset identifier PXD020843. Full blots are shown in Supplementary Information. Source data underlying plots shown in figures are provided in Supplementary Data 1. All other data, if any, are available upon reasonable request.

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

## Acknowledgements
The Yeast two-hybrid screen was performed at the Genomics and Proteomics Core Facility of the German Cancer Research Center (Heidelberg, Germany). We also thank for excellent technical assistance from Katharina Stiebeling and Alexandra Rosskopf. This work is supported by the Else Kröner-Fresenius-Stiftung (EKFS, 2019_A191) and a start-up grant received from medical faculty of the Christian-Albrechts University of Kiel to A.Y.R., and a collaborative research grant from DZHK (German Centre for Cardio-vascular Research) to A.Y.R. and N.F.

## Author contributions
Conceived and designed the experiments: A.Y.R., M.K., A.T., O.J.M., D.F., and N.F.; Performed the experiments and analyzed the data: A.Y.R., A.Bo., N.S., A.D., A.R., M.K., A.Be., L.C., A.H., A.J., and S.S.; Contributed reagents/materials/analysis tools: A.Y.R., A.T., D.F., and N.F.; Wrote the manuscript: A.Y.R., A.R., and N.F.; Revised the manuscript: A.T., O.J.M., D.F., and N.F.

## Funding

## Competing interests
The authors declare no competing interests.
