## [Peer Review File · Communications Biology]

Reviewers' comments:

Reviewer #1 (Remarks to the Author):

In this manuscript, Dr. Frey and colleagues describe a novel role for the E3 ubiquitin ligase HectD3 in the regulation of cardiac hypertrophy. They build on previous work indicating that SUMO2 activates cardiac hypertrophy by identifying HectD3 as an interacting partner for SUMO2. The work is certainly novel, as there are no published reports of functional HectD3 in the heart. They use extensive gain and loss of function *in vivo* and *in vitro* to bolster their claim that HectD3 downregulates SUMO2 and STAT1, collectively conferring antihypertrophic/protective properties. Unfortunately, the inclusion of both SUMO2 and STAT1-related data becomes somewhat confusing and (perhaps as a result) there is a lack of specificity regarding mechanism and causality that detracts from the overall impact of the manuscript in its current form.

MAJOR

1. What evidence do the authors have to support their claim that regulation of STAT1 is critical/causal to the observed effects of HectD3 on hypertrophy and/or inflammation?
2. Are any of the 13 transcripts downregulated by HectD3 overexpression (Figure 3D) bona fide STAT1 targets? Similarly, are the transcripts in Figure 6F STAT1 targets? The authors seem to suggest as much in the final paragraph of their Results section but need to be more specific. As the authors rightly indicate, inflammation in the failing heart is complicated. Do the authors consider cardiomyocyte HectD3 as a regulator of cardiomyocyte cytokine production/immune response? The NRVM data suggest as much. Or are the *in vivo* cytokine data driven to a greater extent by the decrease in macrophage infiltration?
3. What happens to free SUMO2 and sumoylated protein abundance in TAC and AAV-HectD3 mouse hearts? The authors claim in their Discussion "By virtue of post-translationally regulating SUMO2 and its target proteins, HectD3 attenuated the pro-hypertrophic effects of...TAC or Ang II". I do not find data that directly support this claim.
4. The pSTAT1 data in Figure 6D/E are confusing and potentially misleading. It certainly appears that the pSTAT1/tSTAT1 ratios for the HD3 Sham group are among the highest of any group. Yet the authors claim that HectD3 strongly reduces STAT1 phosphorylation (and the summary stats indicate as much). How are the authors normalizing pSTAT1? pSTAT1 should be normalized to tSTAT1 for each sample, rather than comparing pSTAT1 across groups.
5. The overexpression of SUMO2 saturates the capacity for endogenous protein sumoylation. Are high concentrations of free SUMO2 toxic? How do they contribute to the observed effects?
6. How does HectD3 directly target STAT1? It appears that HectD3 overexpression decreases both transcript and protein levels, suggesting a mechanism other than simply ubiquitination.

MINOR

1. Please clarify the significance of the MG132 results in Figures 2E and 2G.
2. Did SUMO2 emerge as a HectD3 target in your TMT labeled proteomics?
3. How do the authors reconcile the discrepant findings regarding the role of SUMO2 in regulating STAT1 phosphorylation in the current study compared to Citation 19?
4. The importance of inflammation in the pathobiology of heart failure is well established. The Discussion would benefit from less consideration of this broad topic and a tighter focus on the data that are presented.

Reviewer #2 (Remarks to the Author):

Summary: This manuscript describes, for the first time, the role of the cardiac ubiquitin ligase HECT domain D3. Specifically, it identifies HECTD3's dual activities of attenuating SUMO2-Calcineurin-NFAT driven cardiomyocyte hypertrophy while abrogating pro-inflammatory actions in the heart. Increasing HECTD3 using AAV9-mediated expression vectors in mice reduced cardiac SUMO2/STAT1 pathological hypertrophy, while also abolishing macrophage infiltration and fibrosis induced by pressure overload.

This outstanding manuscript describes an enormous amount of novel and original data with widespread relevance to pathological cardiac hypertrophy and heart failure. These findings build upon the authors' previous studies of SUMO regulation of nuclear calcineurin/NFAT. This work notably describes for the first time the HECTD3 ubiquitin ligase in the heart, mechanistically dissecting multiple mechanisms in an orderly, convincing, and complete manner. There are only a few minor points that need to be addressed to clarify that an enormous amount of data presented in the manuscript.

Minor points:

1. Studies of the NRVM demonstrate the mechanistic role of HECTD3 in the cardiomyocytes, which parallels the TAC studies of AAV9 HECTD3 treated hearts in vivo. Can the authors provide the expected and observed specificity of the AAV9 expression? AAV9, of course, has tropism for the heart and can be expressed in non-cardiomyocyte cells. Are fibroblasts, inflammatory cells, and/or vasculature expected to express HECTD3? And has this expression pattern been confirmed in vivo? Do other tissues with AAV9 tropism express the AAV9?
2. Expression of the HectD3 in both the nucleus and cytosol with calcineurin and SUMOlated proteins, respectively, is an exciting dichotomy observed. Is there evidence that HectD3's subcellular localization is regulated (e.g., but nuclear import sequences), or are they static independent pools regulating separate proteins?
3. How does pressure overload affect the endogenous HectD3 expression in wildtype hearts? Is HectD3 protein expression level (and subcellular localization) altered in human pathological cardiac hypertrophy (due to stenosis, etc.) and heart failure? Would HectD3 knock-down be expected to exacerbate TAC-related heart failure in vivo? What are the therapeutic implications of the present studies?
4. Since E3 is short for ubiquitin ligase, it seems redundant to use the phrase E3 ubiquitin ligase. In addition, would the authors consider this an E3 SUMO ligase as well? Are there different binding sites within the HECTD3 that bind calcineurin and SUMO-2 to act to SUMOylate and ubiquitinate, respectively?

Reviewers' comments:

Reviewer #1

In this manuscript, Dr. Frey and colleagues describe a novel role for the E3 ubiquitin ligase HectD3 in the regulation of cardiac hypertrophy. They build on previous work indicating that SUMO2 activates cardiac hypertrophy by identifying HectD3 as an interacting partner for SUMO2. The work is certainly novel, as there are no published reports of functional HectD3 in the heart. They use extensive gain and loss of function *in vivo* and *in vitro* to bolster their claim that HectD3 downregulates SUMO2 and STAT1, collectively conferring antihypertrophic/protective properties. Unfortunately, the inclusion of both SUMO2 and STAT1-related data becomes somewhat confusing and (perhaps as a result) there is a lack of specificity regarding mechanism and causality that detracts from the overall impact of the manuscript in its current form.

We thank this reviewer for appreciating our study and providing insightful comments. We understand the critical comment made by the reviewer regarding inclusion of both SUMO2 and STAT1-related data in this manuscript. Nevertheless, we believe that the combined results shown here for SUMO2 and STAT1 complement each other and manifold the impact of HectD3 overexpression as a protective measure against cardiac hypertrophy and inflammation. We however are open to remove SUMO-related data from the manuscript provided the editor and second reviewer also agree to this change.

MAJOR

1. What evidence do the authors have to support their claim that regulation of STAT1 is critical/causal to the observed effects of HectD3 on hypertrophy and/or inflammation?

Both LPS and IFN γ are known to activate i.e. phosphorylate STAT1 thereby inducing a downstream proinflammatory signaling cascade. Indeed, we also observed similar effects of LPS and IFN γ in neonatal rat cardiomyocytes (NRVCMs) (Figure 4A-F). Overexpression of HectD3 not only reduced overall STAT1 levels but also substantially inhibited LPS/IFN γ -mediated STAT1 activation (Figure 4A-F). This led to suppression of inflammatory markers and STAT1 targets like *Bst2*, *Ifit2/3*, *Mx1*, and STAT1 itself when HectD3 was overexpressed in the presence of LPS/IFN γ compared to the respective LacZ controls. In contrast, HectD3 knockdown resulted in increased expression and activation of STAT1 and its downstream targets (Supplementary Figure 4G-I). Moreover, when we used culture supernatant of LPS treated NRVCMs, migration of macrophages was significantly reduced in the culture supernatant of NRVCMs where HectD3 was overexpressed compared to the LacZ control (Figure 6I, 6J). Whereas, use of culture supernatant of NRVCMs where HectD3 was knocked-down resulted in increased macrophage migration (Figure 6I, 6J). Importantly, these *in vitro* effects were also translated *in vivo* where we observed significantly increased levels of STAT1 and p-STAT1 and its transcription targets after TAC, whereas, AAV mediated overexpression of HectD3 strongly attenuated this activation (Figure 6D-F). Furthermore, in line with the *in vitro* findings, HectD3 overexpression led to reduced cardiac infiltration of inflammatory macrophages (Figure 6G, 6H). Finally, we observed that overexpression of STAT1 does not alter cardiomyocyte hypertrophy in NRVCMs (Figure R1A-G). Taken together, these findings support the notion that regulation of STAT1 is critical to the observed beneficial effects of HectD3 on cardiac inflammation.

Figure R1: Effect of STAT1 overexpression on cellular hypertrophy in NRVCMs. A. Immunoblot indicating the Stat1 levels in protein lysate isolated from either control or Stat1 overexpressing NRVCMs. Densitometry for total or phosphorylated STAT1 is presented in box plots **B** and **C**, respectively. **D.** Box plot depicting Stat1 expression at transcript levels determined by quantitative real-time PCR. We observed that the overexpression of Stat1 in NRVCMs did not affect cell surface area in NRVCMs as determined by immunofluorescence microscopy, where cells are stained by α -Actinin and DAPI (**E**) or by measurement of expression levels of natriuretic peptides *Nppa* (**F**) and *NPPB* (**G**) by qRT-PCR.

2. Are any of the 13 transcripts downregulated by HectD3 overexpression (Figure 3D) bona fide STAT1 targets? Similarly, are the transcripts in Figure 6F STAT1 targets? The authors seem to suggest as much in the final paragraph of their Results section but need to be more specific. As the authors rightly indicate, inflammation in the failing heart is complicated. Do the authors consider cardiomyocyte HectD3 as a regulator of cardiomyocyte cytokine production/immune response? The NRVM data suggest as much. Or are the *in vivo* cytokine data driven to a greater extent by the decrease in macrophage infiltration?

We again thank the reviewer for this important observation. Out of 13 dysregulated genes shown in Figure 3D, at least six (*Bst2*, *Ccl2*, *Mx1*, *Pml*, *STAT1* itself and *Tap1*) are known targets of STAT1 (Sato and Tabunoki, *Gene Regul Syst Bio*. 2013; 7: 41–56), whereas, the remaining genes are known targets of other transcription factors involved in interferon responsive signaling. We have now included this information in the results section of revised manuscript. As the reviewer rightly pointed out, based on our *in vitro* results shown in Figure 6F-J and supplementary Figure 6B/C, we indeed believe that - via altering the transcription factor STAT1- HectD3 regulates cardiomyocyte cytokine production (e.g. *CCL2*, *CCL5*) to inhibit infiltration of inflammatory macrophages.

3. What happens to free SUMO2 and sumoylated protein abundance in TAC and AAV-HectD3 mouse hearts? The authors claim in their Discussion “By virtue of post-translationally regulating SUMO2 and its target proteins, HectD3 attenuated the pro-hypertrophic effects of...TAC or Ang II”. I do not find data that directly support this claim.

We would like to point the attention of the reviewer to Supplementary Figure 5A-C where it is observed that the expression of free SUMO2/3 and its sumoylated proteins is significantly higher in the heart of mice after TAC, whereas, AAV9-mediated HectD3 overexpression significantly reduced these levels.

4. The pSTAT1 data in Figure 6D/E are confusing and potentially misleading. It certainly appears that the pSTAT1/tSTAT1 ratios for the HD3 Sham group are among the highest of any group. Yet the authors claim that HectD3 strongly reduces STAT1 phosphorylation (and the summary stats indicate as much). How are the authors normalizing pSTAT1? pSTAT1 should be normalized to tSTAT1 for each sample, rather than comparing pSTAT1 across groups.

Again, we thank the reviewer very much for this important critical observation and comment. Though we completely agree with the reviewer’s views mentioned in this comment, we would like to clarify here the reason why we showed the data for tSTAT1 and pSTAT1 separately. Since we talk about an E3 ubiquitin ligase, HectD3, which acts upon and regulates the protein levels of its substrates via the ubiquitin proteasome system, we trusts that showing the comparative levels of tSTAT1 and pSTAT1 separately will be more suitable for this study than comparing the ratios. For this very reason, we normalized both tSTAT1 and pSTAT1 against Tubulin.

5. The overexpression of SUMO2 saturates the capacity for endogenous protein sumoylation. Are high concentrations of free SUMO2 toxic? How do they contribute to the observed effects?

Thank you very much for raising this interesting issue. As this reviewer would already be aware of the fact that SUMOs interact with their substrates or binding partners not only through covalent attachment i.e. sumoylation but also via a “SUMO interacting motif”, a conserved consensus motif present in some of their target proteins (Proc Natl Acad Sci USA. 2004 Oct 5;101(40):14373-8). Along these lines, we have earlier found that SUMO2 is one of the most robust activators of Calcineurin-NFAT signaling via a sumoylation independent mechanism, where it directly interacts with and tethers Calcineurin to the nucleus thereby causing cardiomyocyte hypertrophy *in vitro* in neonatal rat cardiomyocytes (Sci Rep. 2016 Oct 21;6:35758). Our data also suggests that free SUMO2 and overall SUMO2-mediated sumoylation is significantly increased in human hearts of cardiac hypertrophy patients and experimental animals (Sci Rep. 2016 Oct 21;6:35758). On the other hand, we also found that overexpression of SUMO2 in mouse hearts led to cardiac hypertrophy (Sci Rep. 2016 Oct 21;6:35758). Importantly, all these observed pro-hypertrophic effects of SUMO2 were also observed with the overexpression of sumoylation deficient SUMO2, where the diglycine motif responsible for substrate sumoylation was absent (SUMO2 Δ GG). Overexpression of SUMO2 Δ GG led to induction of comparable hypertrophy to that of wild-type SUMO2 both *in vitro* in neonatal rat cardiomyocytes and *in vivo* in mouse hearts. To further confirm a sumoylation-independent mechanism, we knocked-down UBC9, the only E2-enzyme for SUMO2 which is essential for the ligation of SUMO2 to target proteins (Biochemistry. 2003 Aug 26;42(33):9959-69). We found that the knockdown of UBC9 neither altered the prohypertrophic effects of native SUMO2 nor did it abrogate the increased cellular hypertrophy

mediated by SUMO2 Δ GG in NRVCN (Sci Rep. 2016 Oct 21;6:35758). Taken together, these observations indicate that increased free SUMO2 (which notably also occurs in human heart failure) is indeed toxic, causing cardiac hypertrophy, likely via sumoylation independent mechanism(s), e.g. via tethering Calcineurin to the nucleus.

6. How does HectD3 directly target STAT1? It appears that HectD3 overexpression decreases both transcript and protein levels, suggesting a mechanism other than simply ubiquitination.

Our data shown in Figure 3G indicate that STAT1 is a bona fide substrate of HectD3. Both input and IP blots demonstrate that HectD3 overexpression together with Ubiquitin causes STAT1 ubiquitination. We now also performed an *in vitro* ubiquitination assay and found similar results that STAT1 ubiquitination only occurs when HectD3 is overexpressed with Ubiquitin (Figure R2). This data has now been included in the revised manuscript as Supplementary Figure 3G. Overall, these results indeed support our hypothesis that HectD3 directly targets STAT1 via ubiquitin-proteasome system (UPS).

In regards to alterations in STAT1 transcript levels upon HectD3 overexpression, it is well documented that, once activated, STAT1 can auto-induce its transcript expression since it carries the required binding site (Genes Cells. 2016 Jan;21(1):25-40). We thus postulate that the regulation of the transcript levels of STAT1 by HectD3 overexpression are likely due to the downregulation of activated STAT1.

Figure R2: In vitro ubiquitination assay for STAT1.

MINOR

1. Please clarify the significance of the MG132 results in Figures 2E and 2G.

Thank you again for this suggestion. We have added the following text to the results section concerning Figures 2E and 2G to clarify the significance of the use of MG132.

To determine if the observed effects of HectD3 on SUMO2 are UPS-dependent, we treated NRVCNs with MG132, a potent, reversible, and cell-permeable proteasome inhibitor that results in the accumulation of ubiquitin-conjugated substrate proteins. MG132 treatment attenuated SUMO2 degradation even when HectD3 was overexpressed, supporting the functional importance of this interaction (Figure 2E, 2G).

2. Did SUMO2 emerge as a HectD3 target in your TMT labeled proteomics?

This is an intriguing question. Unfortunately, SUMO2 did not emerge as a HectD3 target in our TMT labeled proteomics analysis. We can only speculate about potential reasons for this finding (unstable binding under these experimental conditions?). However, using immunoblotting, we observed consistently reduced levels of SUMO2/3 and its sumoylated proteins in cardiomyocytes (and also HEK293A cells) when HectD3 was overexpressed.

3. How do the authors reconcile the discrepant findings regarding the role of SUMO2 in regulating STAT1 phosphorylation in the current study compared to Citation 19?

Our interpretation would be that although the basic machinery and the genetic framework are identical in all individual cell types present in an organism, there remains considerable functional and mechanistic cell-type specificity. Based on our consistent observations, we indeed postulate that SUMO2 acts differently on STAT1 in cardiomyocytes compared to e.g. cancer cells likely due to yet unknown or independent mechanism than the one shown by Maarifi et al. (J Immunol. 2015 Sep 1;195(5):2312-24).

4. The importance of inflammation in the pathobiology of heart failure is well established. The Discussion would benefit from less consideration of this broad topic and a tighter focus on the data that are presented.

We have shortened the discussion on the role of inflammation in the pathophysiology of heart failure as per the reviewer's suggestion.

Reviewer #2

Summary: This manuscript describes, for the first time, the role of the cardiac ubiquitin ligase HECT domain D3. Specifically, it identifies HectD3's dual activities of attenuating SUMO2-Calcineurin-NFAT driven cardiomyocyte hypertrophy while abrogating pro-inflammatory actions in the heart. Increasing HectD3 using AAV9-mediated expression vectors in mice reduced cardiac SUMO2/STAT1 pathological hypertrophy, while also abolishing macrophage infiltration and fibrosis induced by pressure overload.

This outstanding manuscript describes an enormous amount of novel and original data with widespread relevance to pathological cardiac hypertrophy and heart failure. These findings build upon the authors' previous studies of SUMO regulation of nuclear calcineurin/NFAT. This work notably describes for the first time the HectD3 ubiquitin ligase in the heart, mechanistically dissecting multiple mechanisms in an orderly, convincing, and complete manner. There are only a few minor points that need to be addressed to clarify that an enormous amount of data presented in the manuscript.

We thank this reviewer for the encouraging words and the appreciation of our study. We also thank this reviewer for some critical observations addressed below and her/his important suggestions.

Minor points:

1. Studies of the NRVM demonstrate the mechanistic role of HectD3 in the cardiomyocytes, which parallels the TAC studies of AAV9 HectD3 treated hearts in vivo. Can the authors provide the expected and observed specificity of the AAV9 expression? AAV9, of course, has tropism for the heart and can be expressed in non-cardiomyocyte cells. Are fibroblasts, inflammatory cells, and/or vasculature expected to express HECTD3? And has this expression pattern been confirmed in vivo? Do other tissues with AAV9 tropism express the AAV9?

This reviewer raises an important issue and of great concern. Indeed AAV9 does transduce other organs such as liver or kidney after systemic vector administration. We thus have chosen to drive HectD3 under transcriptional control of the cardiac-specific human cardiac Troponin-T (hTnT) promoter. As a result, HectD3 expression was only detectable in the heart after intravenous injection of the vector (Figure 5D). In a previous study using a highly sensitive Cre reporter gene, we have confirmed that the hTnT promoter indeed drives expression almost exclusively in cardiomyocytes after intravenous injection of AAV9 vectors (Cardiovasc Res. 2014 Oct 1;104(1):15-23). A significant expression of HECTD3 in cardiac fibroblasts, inflammatory cells or the vasculature is thus highly unlikely. We nevertheless have clarified this issue by adding a detailed description of the vector genome to the material and methods section.

2. Expression of the HectD3 in both the nucleus and cytosol with calcineurin and SUMOlated proteins, respectively, is an exciting dichotomy observed. Is there evidence that HectD3's subcellular localization is regulated (e.g., but nuclear import sequences), or are they static independent pools regulating separate proteins?

We thank this reviewer for this interesting observation. Towards this, using cNLS mapper (Proc Natl Acad Sci USA. 2009 Jun 23;106(25):10171-6; <http://nls-mapper.iab.keio.ac.jp/cgi-bin/NLS Mapper form.cgi>), we found the presence of two potential nuclear localization sequences present in the N-terminal region of human HectD3 with cut-off score of 5.2 and 5.1, respectively (Figure R3). cNLS Mapper extracts putative NLS sequences with a score equal to or more than the selected cut-off score. Higher scores indicate stronger NLS activities. Briefly, a GUS-GFP reporter protein fused to an NLS with a score of 8, 9, or 10 is exclusively localized to the nucleus, that with a score of 7 or 8 partially localized to the nucleus, that with a score of 3, 4, or 5 localized to both the nucleus and the cytoplasm, and that with a score of 1 or 2 localized to the cytoplasm. Cut-off scores of 5.2 and 5.1 identified for HectD3 fits with our observation of presence of HectD3 both in the cytoplasm and the nucleus (Figure 1C and 1G). However, we have not experimentally validated either of these predicted NLS sites. We believe that although it is interesting to understand if and how nuclear localization of HectD3 is regulated and if it has any direct or indirect impact on its cardiac function, it is beyond the scope of current manuscript as it stands as an independent research question. However, using mutagenesis studies, we will further investigate these predicted NLS and this will be one of the important objectives of our future research work on HectD3.

Predicted bipartite NLS		
Pos.	Sequence	Score
21	RFLAEAARSLRAGRPLPAALAFVPREVLYKLYKDP	5.2
93	RDSIELRRGACVVRTTGEELCNGHGLWVKLTKE	5.1

Figure R3: Predicted nuclear localization sequences of human HectD3.

3. How does pressure overload affect the endogenous HectD3 expression in wildtype hearts? Is HectD3 protein expression level (and subcellular localization) altered in human pathological cardiac hypertrophy (due to stenosis, etc.) and heart failure? Would HectD3 knock-down be expected to exacerbate TAC-related heart failure in vivo? What are the therapeutic implications of the present studies?

Thank you again very much for these important comments.

HectD3 expression after TAC: As per reviewer's suggestions, we determined HectD3 transcript and protein levels in wild-type mice after TAC. HectD3 was significantly downregulated at the transcript level in mouse heart after TAC; at protein level however, though we observed a trend of downregulation, no significance was attained (Figure R4). We now also included this data in the revised manuscript as Supplementary Figure 2H-J.

Figure R4: HectD3 protein and transcript levels in mice after TAC.

HectD3 in pathological hypertrophy in human: We would like to highlight here that the data already presented in Figure 2D and 2F shows significant downregulation of HectD3 in human patients suffering of hypertrophic cardiomyopathy. Unfortunately however, we could not perform immunohistology in human patient due to lack of required specimens. Nevertheless, we used TAC vs Sham mice and PE vs PBS treated NRVCMs to find whether stress alters HectD3 localization. Our preliminary assessment suggests no significant differences in either of the conditions, at least by immunofluorescence microscopy. As shown earlier in Figure 1C, HectD3 is present in the cytosol and concentrated towards the perinuclear region and nucleus, whereas, in mouse heart, it is mostly in the cytosol and marginally at perinuclear region and in the nucleus (representative images are shown in Figure R5A, B). In future experiments, we nevertheless plan to carry out subcellular fractionation assays of NRVCMs and mouse heart tissue after pathological stress to know if HectD3 localization is altered.

Figure R5: Immunofluorescence microscopy images of heart tissue (A) and NRVCMs (B) depicting subcellular localization of HectD3.

Cardiac effects after HectD3 knockout: Based on the cardiac protective phenotype against cardiac hypertrophy and inflammation observed in mice upon HectD3 overexpression after pressure overload, we assume that a HectD3 knockdown/knockout would exacerbate TAC-related heart failure *in vivo*. Generation of cardiac-restricted conditional knockout mouse line for HectD3 gene and its characterization for heart function under physiological conditions and pathological stress are part of our prospective studies.

Therapeutic implications of the present study: After promising *in vitro* results in NRVCMs where we found HectD3 to protect cardiomyocytes against hypertrophic or inflammatory insults, we used AAV-mediated gene transfer approach to study its functional impact on the heart in mice *in vivo*. AAV gene therapy vectors are non-hazardous gene transfer shuttles derived from adeno-associated viruses, in which the genetic material of the virus is replaced with therapeutically-useful genetic information. Through the properties of the protein sheath of the original virus, it is possible to use AAV gene therapy vectors for long-term transfer of genetic material into cells of the body, in which important genes are missing or are dysregulated. They also offer the advantage that the gene transfer does not result in the genetic material transferred being incorporated into the genetic material of the cells. Thus, no transition into the germline can take place, and therefore the genetic material introduced cannot be transferred to offspring. Due to all these beneficial effects and the promise AAV-mediated gene therapy delivers, the very idea of using AAVs in our study was to exploit the therapeutic potential of HectD3. Further steps towards determining the promise of treating heart failure using HectD3 overexpression will be to find if these *in vivo* findings can be reproduced in a large animal model of heart failure.

4. Since E3 is short for ubiquitin ligase, it seems redundant to use the phrase E3 ubiquitin ligase. In addition, would the authors consider this an E3 SUMO ligase as well? Are there different binding sites within the HectD3 that bind calcineurin and SUMO-2 to act to SUMOylate and ubiquitinate, respectively?

We thank the reviewer for this comment which indeed is an intriguing question. From the literature, we are aware that several SUMO E3 ligases such as ubiquitin like with PHD and ring finger domains 2 (UHRF2), topoisomerase I binding, arginine/serine-rich (TOPORS), TNF receptor-associated factor 7 (TRAF7), tripartite motif containing 27 (TRIM27), etc. may have dual functions as ubiquitin E3 ligases (J Biol Chem. 2013 Mar 29;288(13):9102-11). HectD3 is an established E3 ubiquitin ligase, and we currently do not know if HectD3 also acts as an E3 SUMO-ligase. Given our findings of regulation of SUMO2 and its sumoylated substrates upon HectD3 overexpression, we speculate that it might act as one, we however do not have any experimental proof at this point supporting this claim.

REVIEWERS' COMMENTS:

Reviewer #1 (Remarks to the Author):

In this revised manuscript, Dr. Frey and colleagues describe a novel role for the E3 ubiquitin ligase HectD3 in the regulation of cardiac hypertrophy. This reviewer greatly appreciates the thorough and thoughtful responses to my initial critiques. At this point, there are really is only one abiding concern:

I appreciate the detailed response to the following query from the initial critique: "What evidence do the authors have to support their claim that regulation of STAT1 is critical/causal to the observed effects of HectD3 on hypertrophy and/or inflammation?" Respectfully, however, they have not actually demonstrated that regulation of STAT1 is critical to HectD3-regulated response. They have provided ample and convincing data that STAT1 levels are inversely proportional to HectD3 expression but there are no data that prove any functional/mechanistic consequence to this phenomenon. Pharmacological or genetic manipulation of STAT1 would be required to do so. If the authors decide against pursuing such experiments, they should soften claims that STAT1 is central to HectD3 biology.

Reviewer #2 (Remarks to the Author):

Minor points: All issues addressed nicely